# Itaconate promotes hepatocellular carcinoma progression by epigenetic induction of CD8+ T-cell exhaustion

Xuemei Gu[1,7], Haoran Wei[2,7], Caixia Suo[3,7], Shengqi Shen[4], Chuxu Zhu[1], Liang Chen[1], Kai Yan[4], Zhikun Li[1], Zhenhua Bian[1], Pinggen Zhang[5], Mengqiu Yuan[5], Yingxuan Yu[1], Jinzhi Du[1], Huafeng Zhang ®[5,6] ✉, Linchong Sun ®[2] ✉ & Ping Gao ®[1,2] ✉

Itaconate is a well-known immunomodulatory metabolite; however, its role in hepatocellular carcinoma (HCC) remains unclear. Here, we find that macrophage-derived itaconate promotes HCC by epigenetic induction of Eomesodermin (EOMES)-mediated CD8+ T-cell exhaustion. Our results show that the knockout of immune-responsive gene 1 (IRG1), responsible for itaconate production, suppresses HCC progression. *Irg1* knockout leads to a decreased proportion of PD-1+ and TIM-3+ CD8+ T cells. Deletion or adoptive transfer of CD8+ T cells shows that IRG1-promoted tumorigenesis depends on CD8+ T-cell exhaustion. Mechanistically, itaconate upregulates PD-1 and TIM-3 expression levels by promoting succinate-dependent H3K4me3 of the *Eomes* promoter. Finally, ibuprofen is found to inhibit HCC progression by targeting IRG1/itaconate-dependent tumor immunoevasion, and high *IRG1* expression in macrophages predicts poor prognosis in HCC patients. Taken together, our results uncover an epigenetic link between itaconate and HCC and suggest that targeting IRG1 or itaconate might be a promising strategy for HCC treatment.

Liver cancer is the third leading cause of cancer death worldwide, with approximately 80% of which being hepatocellular carcinoma (HCC)[1]. While emerging immunotherapy has made significant progress in the treatment of several tumors, including lymphoma and melanoma[2,3], it is not effective for most HCC patients[4,5], largely due to the diverse etiologies of liver cancer and its circumvention of the immune system. Given that the liver is not only an important immune organ but also the largest metabolic organ in the body, it may provide new opportunities for the treatment of HCC from the perspective of immunometabolism.

For decades, the field of immunometabolism has developed significantly, and metabolites traditionally associated with biosynthesis or bioenergetics have been shown to have specific immunomodulatory properties. The immunometabolite itaconate (itaconic acid), the main representative example of metabolic reprogramming in macrophages, has attracted unprecedented attention in recent years[6–8]. It is synthesized from tricarboxylic acid (TCA)-derived cis-aconitate by mitochondrial enzyme immune-responsive gene 1 (IRG1, also known as aconitate decarboxylase 1,

[1]School of Medicine, South China University of Technology, Guangzhou, China. [2]Medical Research Institute, Guangdong Provincial People's Hospital, Guangdong Academy of Medical Sciences, Southern Medical University, Guangzhou, China. [3]Department of Colorectal Surgery, Guangzhou First People's Hospital, School of Medicine, South China University of Technology, Guangzhou, China. [4]Guangdong Cardiovascular Institute, Guangdong Provincial People's Hospital, Guangdong Academy of Medical Sciences, Guangzhou, China. [5]The Chinese Academy of Sciences Key Laboratory of Innate Immunity and Chronic Disease, School of Basic Medical Sciences, Division of Life Science and Medicine, University of Science and Technology of China, Hefei, China. [6]Institute of Health and Medicine, Hefei Comprehensive National Science Center, Hefei, China. [7]These authors contributed equally: Xuemei Gu, Haoran Wei, Caixia Suo. ✉e-mail: hzhang22@ustc.edu.cn; sunlc@mail.ustc.edu.cn; pgao2@ustc.edu.cn

ACOD1). IRG1 is inducible upon macrophage activation with the stimulation of lipopolysaccharide (LPS) and other Toll-like receptor ligands and cytokines, such as type I and type II interferons[9,10]. As a key metabolite in macrophage activation, itaconate is being extensively investigated as an immunomodulator through its ability to activate Nrf2[11], regulate the IκBζ-ATF3 inflammatory axis[12], suppress inflammasome activation by modifying NLRP3 at C548[13], or multiple other mechanisms. Moreover, a series of studies demonstrated that itaconate has bacteriostatic and bactericidal properties by inhibiting the growth of *Salmonella enterica*[14], *Legionella pneumophila*[15,16], and *Staphylococcus aureus*[17]. Oxoglutarate receptor 1 (OXGR1), a G-protein-coupled receptor, is found to specifically sense extracellular itaconate during the pulmonary innate immune response[18]. Recently, myeloid-derived itaconate has been reported to promote the development of melanoma. Specifically, Zhao et al. found that itaconate suppresses the synthesis of aspartate and serine/glycine, thereby attenuating the proliferation and function of CD8+ T cells[19]. All these results indicate that itaconate plays a critical role in various pathological processes. However, with increasing attention being given to the role of itaconate in tumors, the detailed specific molecular mechanisms that might be involved remain to be further clarified.

T cells, especially tumor-specific CD8+ T cells, are the main force that inhibits tumor growth. However, the tumor microenvironment (TME) poses a formidable impediment to T-lymphocyte responses, and various factors within the TME induce T-cell exhaustion and dysfunction[20,21]. The phenotype of exhausted T cells is characterized by gradual loss of robust effector cytotoxicity, reduced cytokine production, upregulation of multiple inhibitory receptors (PD-1, CTLA4, BTLA, TIM-3, and LAG-3), and an altered transcriptional program (*Tbx21*, *Eomes*, *Tox*, *Nfatc1*, *Vhl*, and *Foxo1*)[22–24]. Moreover, exhausted tumor-specific T cells have been reported to coexpress PD-1 and LAG-3[25,26] or PD-1 and TIM-3[27,28]. Competitive metabolism of cancer cells and immune cells leads to a harsh TME[29]. The dysregulation of many nutrients and metabolites in the TME, including glucose, glutamine, arginine, lactate, and succinate, has been proven to play vital roles in tumorigenesis by directly acting on tumor cells or by reshaping the function of tumor-infiltrating lymphocytes (TILs, for example, macrophages, NK cells, and CD8+ T cells)[21,30]. Growing evidence has shown that wholescale epigenetic remodeling is involved in T-cell exhaustion, which prevents T-cell reinvigoration[31]. Nutrient deprivation and deleterious metabolites contribute to metabolism-dependent T-cell exhaustion at the epigenetic level[32]. Itaconate is known to inhibit succinate dehydrogenase complex subunit A (SDHA) activity[33–35], as well as TET DNA dioxygenases that dampen inflammatory responses[36]. However, whether and how metabolic stress, such as itaconate in the TME, affects T-cell exhaustion via epigenetic regulation remains unclear.

In this study, we provide evidence for an unappreciated link between itaconate and HCC. IRG1/itaconate promotes HCC development by inducing the exhaustion of CD8+ T cells in the TME. Mechanistic analysis showed that itaconate promotes the expression of EOMES, which is responsible for the transcriptional regulation of the exhausted markers PD-1 and TIM-3, by accumulating succinate-mediated H3K4me3. Moreover, high expression of *IRG1* in macrophages predicts poor prognosis in clinical HCC patients. Importantly, we found that ibuprofen, a clinical antipyretic analgesic drug, could inhibit HCC by inhibiting IRG1 expression and itaconate production. The combination of ibuprofen and anti-PD-1 antibody shows a better tumor suppressive effect in vivo. In summary, our study demonstrates that abnormal accumulation of IRG1-mediated itaconate production contributes to HCC progression by the epigenetic link between macrophage metabolism and T-cell exhaustion and that ibuprofen can attenuate this process by targeting IRG1 and enhancing the efficacy of PD-1-dependent immunotherapy.

## Results

### Loss of IRG1 suppresses HCC progression in vivo

IRG1 is an emerging metabolic enzyme that is known to be upregulated during macrophage activation, and IRG1-mediated itaconate production plays an effective role in inflammation and host defense[7]. However, the relationship between IRG1/itaconate and HCC is largely unclear. To investigate the role of IRG1/itaconate in tumorigenesis, short-, medium-, and long-term orthotopic HCC models were established by hepatic portal vein injection of Hepa 1-6 cells, intraperitoneal injection of DEN/CCl₄, and hydrodynamic injection of YAP^SSA plasmids in wild-type mice (WT) and *Irg1* knockout mice (KO) (Fig. 1a–c, left panel). Our results showed that the tumorigenesis of KO mice was significantly alleviated compared with that of WT mice in these three HCC models (Fig. 1a–c, middle panel). The liver/body weight ratio (Fig. 1a, b, right panel) and tumor foci (Fig. 1c, right panel) were all obviously decreased in KO mice compared with WT mice. In agreement with the above observation, H&E staining also showed significant elimination of tumor burden in KO mice (Supplementary Fig. 1a–c). We also observed similar results in female mice, although males have a higher probability of developing HCC (Supplementary Fig. 1d). Taken together, these results indicate that IRG1 deficiency obviously suppresses HCC growth, which is sex-independent.

The liver is a frontline immune tissue that is mainly composed of hepatocytes, hepatic stellate cells and immune cells. To explore which group of cells mediates the effect of IRG1 on tumorigenesis, we measured *Irg1* mRNA levels in various hepatic cell subsets. Our quantitative reverse transcriptase PCR (qRT–PCR) results showed that under steady-state conditions without any stimulation, *Irg1* mRNA expression was the highest in macrophages, followed by neutrophils, and low to almost no expression was observed in other cell subsets (Fig. 1d). This result is consistent with the reported high expression of IRG1 in stimulated or activated macrophages[11,33]. By further simulating the tumor microenvironment, we found that IRG1 protein levels and *Irg1* mRNA levels were dramatically increased in macrophages after stimulation with the supernatant of Hepa 1-6 cells (conditional medium 1, CM1) (Supplementary Fig. 1e). Notably, by analyzing the RNA-seq results of HCC patients from TCGA, we found that high expression of *IRG1* in macrophages (CD68^high IRG1^high) in HCC patients predicts poor prognosis compared with low expression of *IRG1* in macrophages (CD68^high IRG1^low), whereas there is no difference in the survival time between HCC patients with CD68^low IRG1^high and CD68^low IRG1^low expression (Fig. 1e). This result prompted us to ask whether macrophage-derived IRG1 contributed to HCC progression. To demonstrate this hypothesis, we neutralized macrophages by using an anti-F4/80 antibody (Supplementary Fig. 1f), and our in vivo results showed that HCC was significantly suppressed in macrophage-depleted WT mice, and there was almost no difference between WT and KO mice after neutralizing macrophages (Fig. 1f). Taken together, these findings indicate that *Irg1* is an oncogene and that macrophage-derived IRG1 promotes HCC development.

### IRG1 drives CD8+ T-cell-mediated tumor immune evasion

The TME is composed of a variety of immune cells, including tumor-associated macrophages (TAMs), myeloid-derived suppressor cells (MDSCs), natural killer cells (NK), neutrophils, and T cells. Changes in cell number, subpopulation, and activation state can affect the immune responsiveness of immune cells to surrounding tumor cells. Since Fig. 1f indicates that macrophages played an important role in HCC development promoted by IRG1, we performed t-distributed stochastic neighbor embedding (t-SNE) analysis of livers on YAP^SSA-induced mouse HCC samples to dissect which groups of immune cells are involved in IRG1 regulation of HCC. Interestingly, we found that the number of tumor-infiltrating PD-1+ CD8+ T cells in the livers of KO mice was significantly reduced (Fig. 2a and Supplementary Fig. 2a). This result suggests that IRG1-mediated HCC progression may depend on

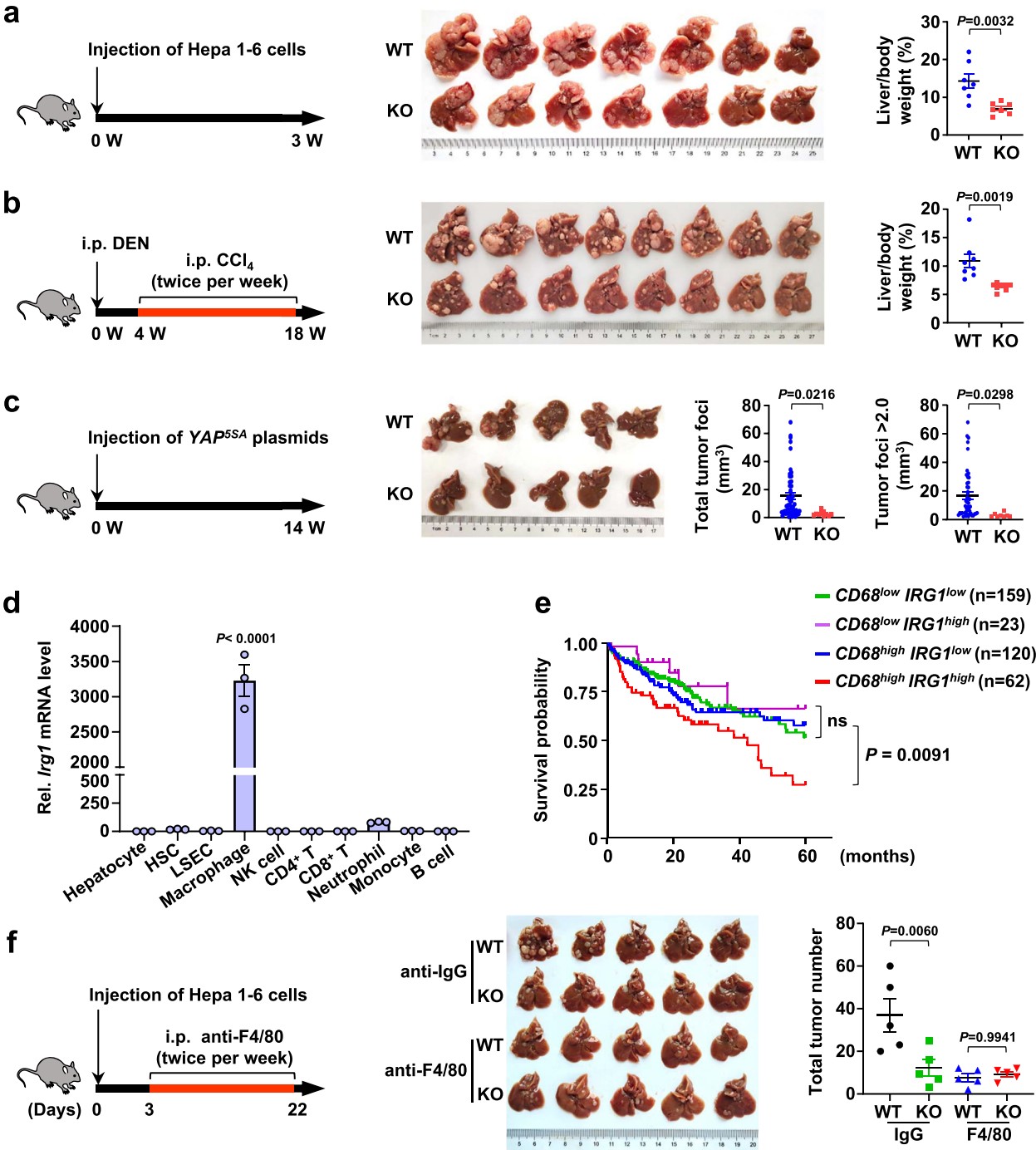

**Fig. 1 | Loss of IRG1 suppresses HCC progression in vivo. a** Schematic diagram of the Hepa 1-6 cell-induced HCC model (left panel). C57BL/6 wild-type (hereafter referred to as WT) and $Irg1^{-/-}$ (hereafter referred to as KO) mice were injected with Hepa 1-6 cells through the hepatic portal vein. Representative liver images are shown (middle panel), and the ratio of liver/body weight was determined (right panel). $n = 7$ mice per group. **b** Schematic diagram of the DEN/CCl$_4$-induced HCC model (left panel). WT and KO mice were intraperitoneally (i.p.) injected with DEN, and 4 weeks later, they were injected with CCl$_4$ twice per week for 14 weeks. Representative liver images are shown (middle panel), and the ratio of liver/body weight was determined (right panel). $n = 8$ mice per group. **c** Schematic diagram of the $YAP^{5SA}$-induced HCC model (left panel). Plasmids expressing $YAP^{5SA}$ together with plasmids expressing the Sleeping Beauty transposase (PB) were delivered into WT and KO mice by hydrodynamic injection. Representative liver images are shown (middle panel), and total tumor foci and tumors larger than 2.0 mm$^3$ were

measured (right panel). $n = 5$ mice per group. **d** $Irg1$ mRNA levels were measured in nonimmune cells and immune cells isolated from the livers of WT mice. $n = 3$ mice per sample. **e** Kaplan–Meier analysis of HCC patient survival was performed based on $CD68$ and $IRG1$ mRNA levels. The patient liver tissue RNA-seq was divided into $CD68^{low}$ $IRG1^{low}$, $CD68^{low}$ $IRG1^{high}$, $CD68^{high}$ $IRG1^{low}$, and $CD68^{high}$ $IRG1^{high}$. **f** WT or KO mice were i.p. injected with anti-F4/80 or control anti-IgG2b antibody twice per week for 3 weeks after receiving Hepa 1-6 tumor cell injection (left panel). Representative liver images are shown (middle panel), and liver surface tumor nodules were counted (right panel). $n = 5$ mice per group. All data represent mean ± SEM. Statistical significance was determined by unpaired two-tailed Student's $t$-test (**a**, **b**, **c**), one-way ANOVA (**d**), Log-rank (Mantel-Cox) test (**e**) and two-way ANOVA with Tukey's correction (**f**). Data are representative of three independent experiments with similar results (**d**). Source data are provided as a Source Data file.

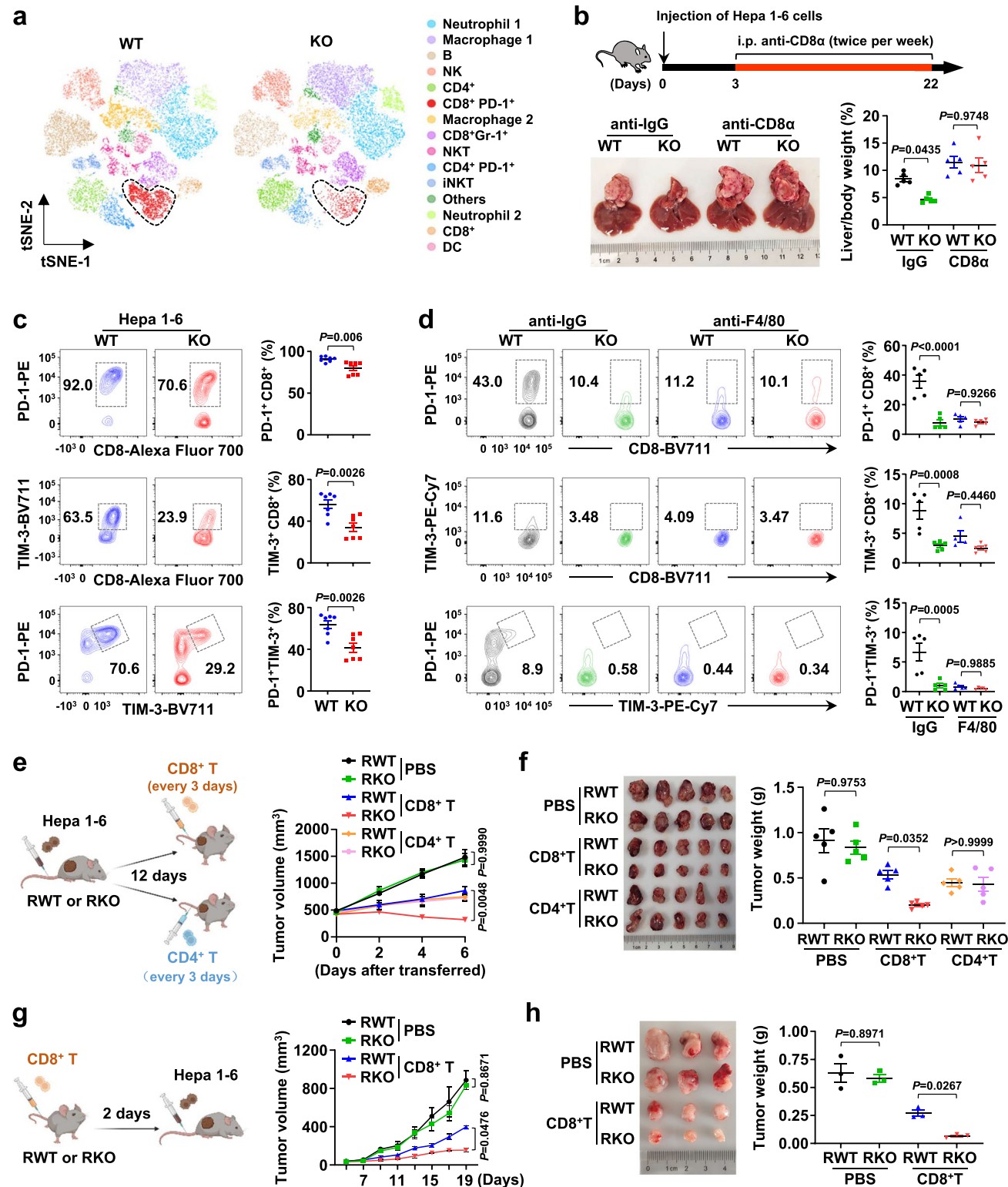

CD8[+] T cells. To test this hypothesis, we used an anti-CD8α antibody to neutralize CD8[+] T cells during the progression of Hepa 1-6 cell-induced orthotopic HCC. Our results showed that there was almost no difference in HCC between WT and KO mice under anti-CD8α antibody treatment (Fig. 2b and Supplementary Fig. 2b, c).

In the TME of HCC, melanoma, lung cancer and other tumors, CD8[+] T cells differentiate into an exhausted status, which becomes a barrier to effective antitumor immunity. Exhausted CD8[+] T cells highly express inhibitory receptors (PD-1, TIM-3, etc.) and

transcription factors such as EOMES and BATF but produce low levels of cytotoxic molecules (TNF, IFN-γ, etc.)[37,38]. We analyzed the expression of the exhaustion markers PD-1 and TIM-3 in tumor-infiltrating CD8[+] T cells in Hepa 1-6 cell-induced (Fig. 2c), DEN/CCl4-induced (Supplementary Fig. 2d), and *YAP^SSA*-induced HCC models (Supplementary Fig. 2e) and found that PD-1[+] CD8[+], TIM-3[+] CD8[+], and PD-1[+] TIM-3[+] CD8[+] T cells were all significantly reduced in KO mice. We next examined and found that the positive staining of the cytotoxic molecules IFN-γ and TNF (typical markers for effector T-cell

**Fig. 2 | IRG1 drives CD8⁺ T-cell-mediated tumor immune evasion. a** The t-Distributed Stochastic Neighbor Embedding (t-SNE) plot shows the projection of various immune cells in livers from the $YAP^{S5A}$-induced liver cancer model of WT and KO mice. The functional description of each cluster shown in different colors is determined by the gene expression characteristics of each cluster. $n = 5$ mice per group. **b** WT or KO mice were i.p. injected with anti-CD8α or control anti-IgG2b antibody for 3 weeks after receiving Hepa 1-6 tumor cell injection (upper panel). Representative liver images are shown, and the ratio of liver/body weight was measured (lower panel). $n = 5$ mice per group. **c** Representative flow cytometry data and summary plot of the frequency showing the percentage of PD-1⁺, TIM-3⁺, and PD-1⁺ TIM-3⁺ cells among CD8⁺ TILs isolated from livers of WT and KO mice with Hepa 1-6 cell-induced liver cancer. $n = 7$ mice per group. **d** Representative flow cytometry data and summary plot of the frequency showing the percentage of PD-1⁺, TIM-3⁺, and PD-1⁺ TIM-3⁺ cells among CD8⁺ TILs isolated from livers of WT and KO mice with Hepa 1-6 cell-induced liver cancer that were i.p. injected with anti-F4/80 or control anti-IgG2b antibody. $n = 5$ mice per group. **e** CD8⁺ T cells or CD4⁺ T cells isolated from the spleen of WT mice were transferred into $Irg1^{+/+}$ $Rag^{-/-}$ (referred to as RWT) and $Irg1^{-/-}$ $Rag^{-/-}$ (referred to as RKO) mice from Day 12 post-tumor implantation. Schematic diagram, tumor volume, and **f** tumor images are shown, and tumor masses were measured. $n = 5$ mice per group. **g** CD8⁺ T cells isolated from the spleens of WT mice were transferred into RWT and RKO mice through i.v. injection. Two days later, Hepa 1-6 cells were injected into RWT and RKO mice. Schematic diagram, tumor volume, and **h** tumor images are shown, and tumor masses were measured. $n = 3$ mice per group. All data represent mean ± SEM. Statistical significance was determined by unpaired two-tailed Student's $t$-test (**c**) and two-way ANOVA with Tukey's correction (**b**, **d**–**h**). Source data are provided as a Source Data file.

persistence) in tumor-infiltrating CD8⁺ T cells were increased in KO mice compared with WT mice (Supplementary Fig. 2f). This result further confirms that IRG1 induces an exhausted state in tumor-infiltrating CD8⁺ T cells.

Macrophages are extensively linked with exhausted T cells in the TME[39]. As expected, PD-1⁺ CD8⁺ and TIM-3⁺ CD8⁺ tumor-infiltrating T cells were significantly reduced in WT mice after macrophage depletion (Fig. 2d). Interestingly, the percentage of PD-1⁺ CD8⁺ and TIM-3⁺ CD8⁺ T cells displayed little difference between WT and KO mice under anti-F4/80 antibody treatment for HCC (Fig. 2d). To further dissect the contribution of CD8⁺ T cells to IRG1-promoted HCC tumorigenesis, $Rag^{-/-}$ mice, which lack mature T cells and B cells, were cross mated with $Irg1^{-/-}$ mice to obtain $Irg1^{+/+}$ $Rag^{-/-}$ (RWT) and $Irg1^{-/-}$ $Rag^{-/-}$ (RKO) mice. CD4⁺ T cells and CD8⁺ T cells isolated from WT mice were adoptively transferred to RWT and RKO mice on Day 12 of tumor implantation (Fig. 2e). As expected, there were no differences between the two groups of mice treated with control PBS, and CD8⁺ T-cell transfer significantly suppressed tumor growth in the RKO group compared with the RWT group. Notably, we found that although the CD4⁺ T-cell-transferred groups exhibited suppressed tumor volume and tumor weight, there was no significant difference between these two groups (Fig. 2e, f). This finding suggests that CD8⁺ T cells, rather than CD4⁺ T cells, are the main factor involved in IRG1-mediated tumorigenesis. Furthermore, CD8⁺ T cells were adoptively transferred to RWT and RKO mice 2 days before tumor implantation. The results showed that there was no difference in tumor volume or tumor weight between the two groups of mice treated with control PBS, and CD8⁺ T-cell transfer significantly suppressed tumor growth in the RKO group compared with the RWT group (Fig. 2g, h). Consistently, PD-1⁺ CD8⁺, TIM-3⁺ CD8⁺, and PD-1⁺ TIM-3⁺ CD8⁺ T cells were significantly reduced in the peripheral blood of RKO mice after CD8⁺ T-cell transfer, as shown in Fig. 2g (Supplementary Fig. 2g). Together, these results suggest that IRG1-promoted HCC development depends on the exhaustion of tumor-infiltrating CD8⁺ T cells.

**Macrophage-derived itaconate induces CD8⁺ T-cell exhaustion**
IRG1 is mainly expressed in macrophages (Fig. 1d) and catalyzes the generation of itaconate from cis-aconitate[7]. However, whether itaconate mediates IRG1-promoted T-cell exhaustion remains unknown. Elevated intracellular itaconate levels have been reported in mouse bone marrow-derived macrophages (BMDMs) after LPS stimulation[11]. Here, we found that itaconate was detected not only intracellularly but also extracellularly in both LPS-stimulated (12 h) and CM1-treated (24 h) BMDMs isolated from WT mice but not in KO mouse-derived BMDMs treated accordingly (Fig. 3a). These results indicate that itaconate could be secreted into the TME, which prompted us to ask whether macrophage-derived itaconate promotes tumor-infiltrating T-cell exhaustion and tumorigenesis. To test this hypothesis, WT and KO mice were treated with the cell-permeable itaconate derivative 4-octyl itaconate (4-OI) (25 mg/kg per mouse), which has been shown to be released intracellularly in the form of itaconate[11]. Accordingly, increased itaconate abundance was confirmed in the liver tissues of KO mice treated with 4-OI by UPLC–MS/MS (Supplementary Fig. 3a). WT and KO mice treated with 4-OI displayed more severe tumors than those treated with control dimethyl sulfoxide (DMSO), and there was no difference between WT and KO mice after 4-OI treatment (Fig. 3b and Supplementary Fig. 3b). Similar results were observed for the proportion of terminally exhausted tumor-infiltrating T-cell subsets, including PD-1⁺ CD8⁺, TIM-3⁺ CD8⁺, and PD-1⁺ TIM-3⁺ CD8⁺ T cells (Fig. 3c). Additionally, 4-OI treatment reduced the percentage of TNF⁺ CD8⁺ and IFN-γ⁺ CD8⁺ tumor-infiltrating T cells in WT mice and led to no difference in the cytotoxic capacity of CD8⁺ T cells between WT and KO mice (Supplementary Fig. 3c). Consistently, 4-OI treatment could also promote liver cancer progression and diminished the difference between WT and KO mice in the $YAP^{S5A}$-induced HCC model (Supplementary Fig. 3d). These results thus indicate that IRG1-mediated tumor-infiltrating T-cell exhaustion and tumorigenesis depend on itaconate in vivo.

Itaconate is generated mainly by macrophages, and macrophages are extensively linked to T-cell exhaustion[39]. To test whether macrophage-derived endogenous itaconate directly promotes T-cell exhaustion, CD8⁺ T cells were cocultured with BMDMs in a Transwell chamber. Consistent with the in vivo results, the proportions of PD-1⁺ CD8⁺, TIM-3⁺ CD8⁺, and PD-1⁺ TIM-3⁺ CD8⁺ T-cell subsets cocultured with KO-derived BMDMs were significantly reduced compared with those cocultured with WT-derived BMDMs (Fig. 3d). In addition, after repeated activation in the presence of 4-OI, CD8⁺ T cells also showed an enhanced proportion of the exhausted subset denoted by PD-1⁺ and TIM-3⁺ (Supplementary Fig. 3e).

To further investigate whether the tumor cell-killing ability of CD8⁺ T cells depends on itaconate, activated CD8⁺ T cells were treated with 4-OI and then were cocultured with Hepa 1-6 cells. Along with the upregulation of the exhaustion markers PD-1 and TIM-3 (Fig. 3e), higher numbers of Hepa 1-6 cells were detected in the 4-OI-treated group of CD8⁺ T cells (Fig. 3e). More interestingly, the addition of 4-OI did not directly promote the proliferation of Hepa 1-6 cells (Supplementary Fig. 3f). Hepatic macrophage populations are distinguished into functionally different subsets such as resident Kupffer cells and monocytes-derived macrophages (MoMs). We then sorted F4/80⁺ cells, Kupffer cells, and MoMs from livers with Hepa 1-6 cell-induced tumor and observed no significant variation in the expression of $Irg1$ among these cells (Supplementary Fig. 3g, h). By analyzing the percentages of PD-1⁺ and TIM-3⁺ T cells, we observed no significant difference between Kupffer cells and MoMs in itaconate-mediated induction of CD8⁺ T-cell exhaustion compared to F4/80⁺ cells (Supplementary Fig. 3i). Collectively, these results suggest that IRG1-mediated T-cell exhaustion and tumorigenesis depend on itaconate levels both in vitro and in vivo and that macrophage-derived itaconate indirectly promotes tumor growth by suppressing CD8⁺ T-cell activity.

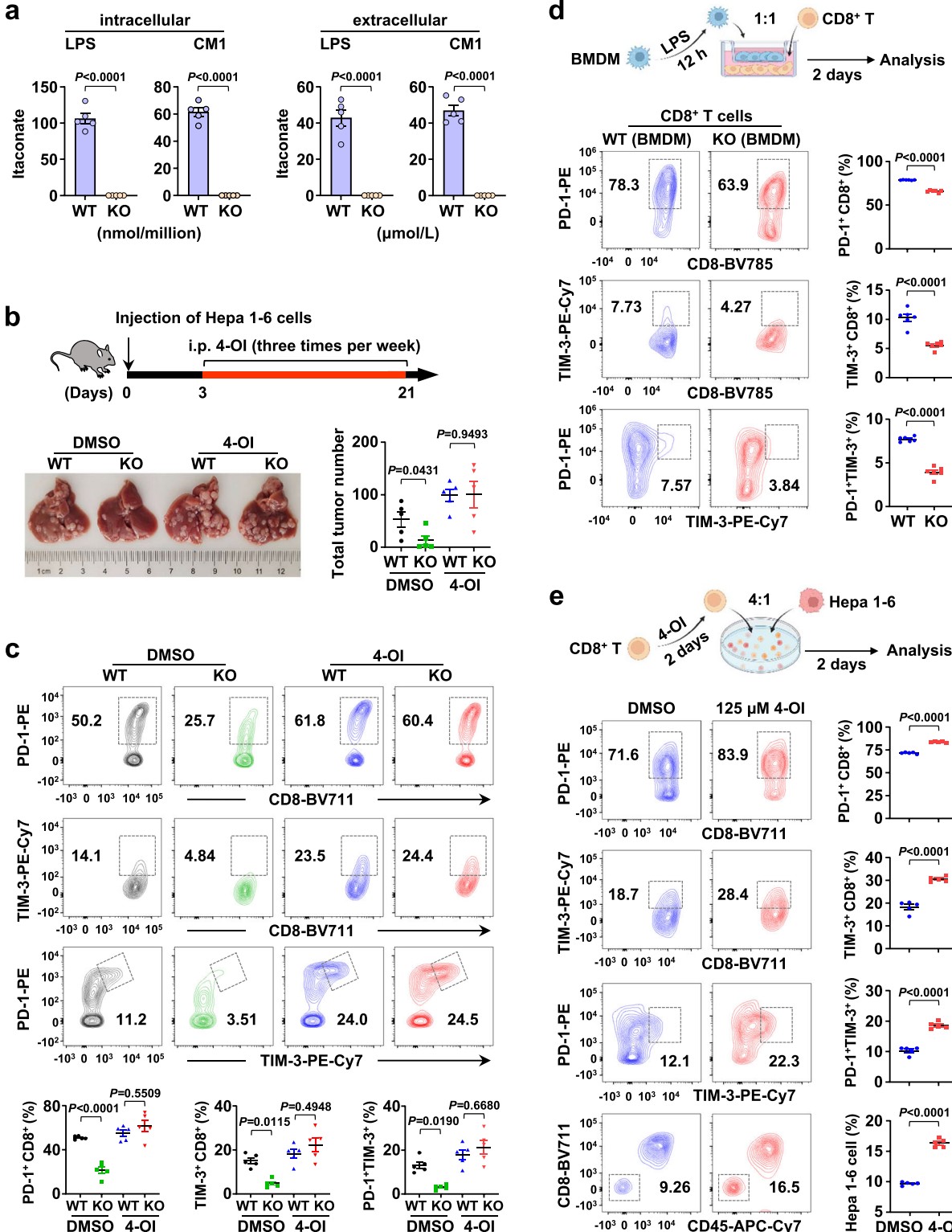

## Itaconate induces CD8+ T-cell exhaustion by promoting succinate-mediated H3K4me3 of *Eomes*

Next, we sought to determine how IRG1-mediated itaconate production promotes CD8+ T-cell exhaustion. As hallmarks of exhausted CD8+ T cells, PD-1 and TIM-3 are regulated by a number of exhaustion-related transcription factors (TFs) (*IRF4, EOMES, YY1, PRDM1,* etc.)[31,32]. To investigate which TFs are responsible for PD-1 and TIM-3 expression during IRG1-induced CD8+ T-cell exhaustion, we examined the mRNA levels of exhaustion-related TFs in (1) CD8+ T cells isolated from the

livers of Hepa 1-6 cell-induced HCC of WT and KO mice; (2) Jurkat cells treated with 4-OI for 48 h; and (3) CD8+ T cells isolated from the livers of WT and KO mice treated with 4-OI during HCC induction (Supplementary Fig. 4a). By combining the above three screening results, we found that the changes in *Eomes* were consistent (Fig. 4a). EOMES, a T-box transcription factor, is known to regulate CD8+ T-cell exhaustion. The ChIP-seq results of EOMES-overexpressing T cells revealed that EOMES directly controls the expression of T-cell exhaustion genes, such as *Pdcd1* (encoding PD-1) and *Havcr2* (encoding TIM-3)[37].

**Fig. 3 | Macrophage-derived itaconate induces CD8⁺ T-cell exhaustion. a** UPLC–MS/MS analysis of the itaconate abundance in the supernatant and intracellular fraction of WT- and KO-derived BMDMs stimulated by LPS (100 ng/ml, 12 h) and CM1 (1:1, 24 h). $n = 5$ samples per group. **b** WT or KO mice were i.p. treated with control DMSO or 4-OI three times per week for 3 weeks after receiving Hepa 1-6 tumor cell injection (upper panel). Representative liver images are shown, and liver surface tumor nodules were counted (lower panel). $n = 5$ mice per group. **c** Representative flow cytometry data (upper panel) and summary plot of the frequency (lower panel) showing the percentage of PD-1⁺, TIM-3⁺, and PD-1⁺ TIM-3⁺ cells among CD8⁺ TILs isolated from the livers in (**b**). $n = 5$ mice per group. **d** BMDMs isolated from WT and KO mice were cultured with L929 cell supernatant and then stimulated with 100 ng/ml LPS for 12 h. BMDMs were cocultured with CD8⁺ T cells at a proportion of 1:1 for 2 days. Representative flow cytometry data and summary plot showing the percentages of PD-1⁺, TIM-3⁺, and PD-1⁺ TIM-3⁺ cells among CD8⁺ T cells. **e** Hepa 1-6 cells and CD8⁺ T cells were cocultured at a ratio of 1:4 for 2 days. Before coculture, CD8⁺ T cells were treated with 4-OI (125 μM) for 48 h. Representative flow cytometry data and summary plot showing the percentages of PD-1⁺, TIM-3⁺, and PD-1⁺ TIM-3⁺ cells among CD8⁺ T cells. The percentage of Hepa 1-6 cells was defined as CD8⁻ CD45.2⁻ in the coculture system. All data represent mean ± SEM. Statistical significance was determined by unpaired two-tailed Student's $t$-test (**a, b, d, e**) and two-way ANOVA with Tukey's correction (**c**). Data are representative of three independent experiments with similar results (**d, e**). Source data are provided as a Source Data file.

By coculturing CD8⁺ T cells with BMDMs, the EOMES⁺ as well as EOMES⁺ PD-1⁺ and EOMES⁺ TIM-3⁺ CD8⁺ T-cell subsets were obviously reduced when cocultured with KO-derived BMDMs (Fig. 4b). Consistently, *Eomes* mRNA levels were decreased in CD8⁺ T cells from YAP^5SA-induced KO mouse livers (Supplementary Fig. 4b). Meanwhile, EOMES⁺ CD8⁺, PD-1⁺ CD8⁺, and TIM-3⁺ CD8⁺ T cells were also significantly reduced in the liver tissues of female KO mice, as shown in Supplementary Fig. 1d (Supplementary Fig. 4c).

Widespread epigenetic alteration is known to regulate CD8⁺ T-cell exhaustion. Itaconate is known to cause succinate accumulation by inhibiting succinate dehydrogenase (SDH) activity[33], and our previous work showed that succinate regulates histone modification in tumor cells[40]. Decreased succinate levels were observed in KO-derived BMDMs treated with LPS or CM1 compared with WT-derived BMDMs (Supplementary Fig. 4d). Consistent with the phenomenon in macrophages, 4-OI treatment led to the accumulation of succinate accompanied by increased α-KG levels but decreased fumarate levels in CD8⁺ T cells (Fig. 4c). More importantly, the increased succinate/α-KG ratio and decreased fumarate/α-KG ratio upon 4-OI addition indicate suppressed SDH activity in converting succinate to fumarate in CD8⁺ T cells (Fig. 4c). These results showed that itaconate inhibits SDH activity and promotes succinate accumulation in CD8⁺ T cells.

Previous reports have shown that accumulated cellular succinate regulates epigenetic reprogramming by suppressing JHDM activity[41]. We thus detected histone methylation levels in 4-OI-treated Jurkat cells. Our results showed that the H3K4me3 level was dramatically increased among the histone methylation modifications (Fig. 4d). To further investigate the relationship between H3K4me3 and *Eomes*, we treated Jurkat cells and CD8⁺ T cells with 4-OI and found obvious binding of H3K4me3 to the potential binding site in the *Eomes* gene promoter region by ChIP–PCR (Fig. 4e). To further examine whether succinate accumulation regulates the H3K4me3-*Eomes*-*Pdcd1*/*Havcr2* axis, we treated Jurkat cells with dimethyl succinate (DMS, a membrane-permeable succinate analog) and dimethyl malonate (DMM, an inhibitor of SDH) to detect H3K4me3 and EOMES protein levels, as well as *PDCD1* and *HAVCR2* mRNA levels. Our results showed that both DMS and DMM treatment increased H3K4me3 and EOMES protein levels and promoted the transcription of *PDCD1* and *HAVCR2* (Fig. 4f, g). Similar results were also observed in both DMS- and DMM-treated CD8⁺ T cells (Supplementary Fig. 4e).

Although there is currently no available report on the mechanism by which T cells internalize itaconate, given the similar chemical structure of itaconate and succinate (Supplementary Fig. 4f), we speculate that itaconate may shuttle into the interior of the cells through the same membrane carrier monocarboxylate transporter 1 (MCT1) utilized by succinate. Three compounds, namely 7ACC2, AZD3965, and BAY-8002, have been reported as inhibitors of MCT1 to impede the entry of succinate into cells[42]. Meanwhile, succinate can be absorbed by T cells via MCT1[43]. Furthermore, our simulation shows that both itaconate and succinate bind to the substrate pocket of MCT1 (Supplementary Fig. 4f), which indicates that MCT1 can transport itaconate into cells. Those lines of evidence suggest that T cells may

internalize itaconate via MCT1. To test this hypothesis, CD8⁺ T cells from WT mice were treated with MCT1 inhibitors before itaconate treatment. The results showed that intracellular itaconate is indeed partially reduced when treating CD8⁺ T with MCT1 inhibitors (Supplementary Fig. 4g). Taken together, these results suggested that MCT1 is a potential transporter that mediates the internalization of itaconate to CD8⁺ T cells.

Altogether, these results showed that macrophage-derived itaconate suppresses SDH activity and promotes succinate accumulation in CD8⁺ T cells, leading to H3K4me3-mediated *Eomes* transcription and the subsequent upregulation of the exhaustion markers PD-1 and TIM-3 (Fig. 4h).

## Ibuprofen inhibits HCC by blocking IRG1/itaconate-regulated immune evasion

Our above results suggest that IRG1/itaconate plays oncogenic roles in HCC tumorigenesis. Considering its excessive association with inflammation, we wondered whether clinical inflammation-related drugs could be used to treat HCC by targeting IRG1. We thus screened clinical analgesic-antipyretic and anti-hepatitis drugs, and our results showed that ibuprofen, bifendatatum, ribavirin, etc., could suppress *Irg1* mRNA levels in RAW264.7 cells after LPS treatment (Supplementary Fig. 5a). However, the immunoblotting results showed that only ibuprofen consistently and effectively inhibited IRG1 in both RAW264.7 cells and BMDMs (Supplementary Fig. 5b), and ibuprofen had a dose-dependent inhibitory effect on *Irg1* mRNA and IRG1 protein levels (Supplementary Fig. 5c). Previous studies have reported that anti-inflammatory agent ibuprofen showed inhibitory effect on nuclear transcription factor-kappa B (NF-κB)[44,45], and knockout of p50 (NF-κB subunit) reduced the expression of *Irg1* in BMDMs[46]. To investigate the relationship between NF-κB (p50) and *Irg1*, we treated BMDMs with ibuprofen and found attenuated binding activity of p50 to the promoter of the *Irg1* gene by ChIP–PCR (Supplementary Fig. 5d). Furthermore, Hepa 1-6-derived supernatant (CM1)-induced IRG1 upregulation was suppressed after ibuprofen treatment in RAW264.7 cells (Fig. 5a). Consistently, after coculture with ibuprofen-treated RAW264.7-derived supernatant (CM4), the increase in the percentage of EOMES⁺ CD8⁺, PD-1⁺ CD8⁺, and TIM-3⁺ CD8⁺ T cells and *Eomes*, *Pdcd1* and *Havcr2* mRNA levels in CD8⁺ T cells induced by RAW264.7 cell-derived supernatant (CM3 vs. CM2) were reversed (Fig. 5b and Supplementary Fig. 5e).

Ibuprofen is widely used to relieve pain; however, its role in HCC development is not clear. Our in vivo experiments showed that ibuprofen has a dose-dependent inhibitory effect on Hepa 1-6-derived allograft growth. Low-dose ibuprofen (20 mg/kg) showed obvious antitumor effects. Although high-dose ibuprofen (80 mg/kg) was more effective in tumor inhibition, several mice died due to side effects (Supplementary Fig. 5f, g). We then used 20 mg/kg ibuprofen in the Hepa 1-6 cell-induced orthotopic HCC model and found that low-dose ibuprofen dramatically suppressed tumor growth (Fig. 5c, d). Interestingly, exogenous 4-OI addition restored ibuprofen-inhibited tumor growth, suggesting that ibuprofen-inhibited tumor growth was

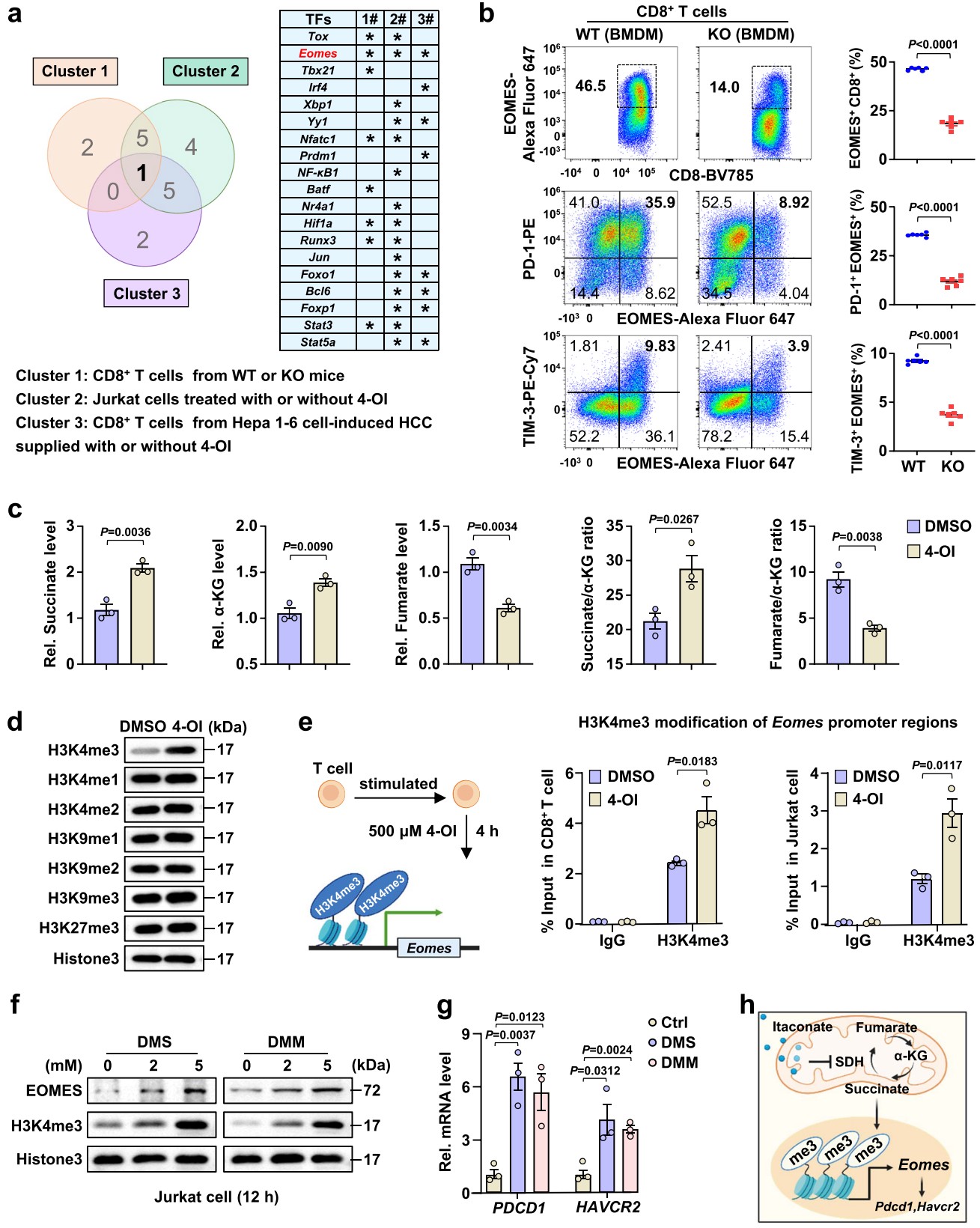

dependent on itaconate (Fig. 5c, d and Supplementary Fig. 5h). Furthermore, itaconate levels were significantly decreased in the liver tissue of ibuprofen-treated mice compared with controls, while the level of itaconate recovered after the addition of 4-OI in the ibuprofen-treated group (Fig. 5e). Meanwhile, ibuprofen treatment reduced the percentage of tumor-infiltrating EOMES⁺ CD8⁺, PD-1⁺ CD8⁺, and TIM-3⁺

CD8⁺ T cells, which was reversed by 4-OI treatment (Fig. 5f). These results suggest that ibuprofen inhibits the development of HCC by blocking itaconate production.

Emerging immunotherapies have made significant advances in several tumors but are not particularly effective in liver cancer patients. Based on our finding that ibuprofen inhibits the production

**Fig. 4 | Itaconate induces CD8+ T-cell exhaustion by promoting succinate-mediated H3K4me3 of *Eomes*. a** Venn diagram of three separate experiments: exhausted TFs were detected in CD8+ T cells isolated from mouse liver or Jurkat cells. **b** Representative flow cytometry data and summary plot showing the percentages of EOMES+, EOMES+ PD-1+, and EOMES+ TIM-3+ cells among CD8+ T cells from Fig. 3d. **c** Measurement of intracellular succinate, α-KG, fumarate, succinate/α-KG and fumarate/α-KG in CD8+ T cells treated with DMSO or 4-OI by an assay kit. **d** Immunoblotting analysis of the protein levels of H3K4me1/2/3, H3K9me1/2/3, and H3K27me3 in Jurkat cells stimulated with PHA and further treated with DMSO or OI. The experiments were repeated three times independently with similar results. **e** ChIP experiments were performed in CD8+ T cells isolated from the spleens of WT

mice or Jurkat cells treated with DMSO or 4-OI using IgG or H3K4me3 antibodies. Jurkat cells were stimulated with PHA before 4-OI treatment. The occupancy of potential binding sites in the *Eomes* gene by H3K4me3 was determined by qRT–PCR. **f** Immunoblotting analysis of the nuclear protein levels of H3K4me3 and EOMES in Jurkat cells stimulated with PHA in the presence of DMS or DMM. **g** Analysis of the mRNA levels of *PDCD1* and *HAVCR2* in Jurkat cells stimulated with PHA in the presence of DMS or DMM. **h** Diagram of the itaconate-succinate-H3K4me3-*Eomes* axis in CD8+ T cells. All data represent mean ± SEM. Statistical significance was determined by unpaired two-tailed Student's *t*-test (**b**, **c**, **e**, **g**). Data are representative of three independent experiments with similar results (**b**–**g**). Source data are provided as a Source Data file.

---

of itaconate, which promotes T-cell exhaustion, we next aimed to investigate whether ibuprofen enhances immunotherapy efficiency. A dose-dependent suppressive effect on allograft growth was observed by administering anti-PD-1 antibody to mice (Supplementary Fig. 5i, j). Since the tumor inhibition effect of low-dose anti-PD-1 antibody (20 µg per mouse) was lower than that of high-dose anti-PD-1 antibody (100 µg per mouse) and ibuprofen could inhibit HCC by suppressing itaconate-mediated T-cell exhaustion, we subsequently investigated whether the combination of low-dose anti-PD-1 antibody and low-dose ibuprofen has a synergistic inhibitory effect on HCC progression. Interestingly, our combination experiments in vivo showed that ibuprofen dramatically enhanced the antitumor effect of the anti-PD-1 antibody in both intrahepatic tumor model and subcutaneous allograft tumor model (Fig. 5g–i and Supplementary Fig. 5k, l). In summary, these findings suggest that the clinical analgesic-antipyretic drug ibuprofen could suppress HCC by targeting IRG1/itaconate-mediated T-cell exhaustion, and ibuprofen significantly improved the efficacy of anti-PD-1 antibody-based immunotherapy.

## Discussion

The TME has long been described as a "metabolic wasteland" because of the lack of glucose, amino acids, oxygen, and other essential nutrients, which leads to starvation and dysfunction of infiltrating immune cells. However, this view is one-sided. The byproducts of nutrient consumption, such as lactic acid, arginine, tryptophan, reactive oxygen species (ROS) and adenosine, play important roles in shaping immune cell function and thus should not be ignored in cancer immunotherapy[47]. Itaconate has emerged in recent years as a metabolite that is mainly synthesized in macrophages and has macrophage-related functions[48–50]. It has been extensively studied based on its immunomodulatory activity. Here, we uncovered how itaconate accelerates HCC development by inducing the exhaustion of intratumoral CD8+ T cells. Itaconate activated the expression of PD-1 and TIM-3 in CD8+ T cells at the epigenetic level through the succinate-H3K4me3-*Eomes* signaling pathway, which could be suppressed by ibuprofen, a clinical drug that was redefined as a potent inhibitor of IRG1 in this study. Importantly, HCC patients with high *IRG1* expression in macrophages had poorer prognoses, underscoring the clinical relevance of our findings.

Myeloid-derived suppressor cells (MDSC)-derived itaconate has been shown to play a cancer-promoting role by suppressing the biosynthesis of aspartate and serine/glycine in CD8+ T cells[19]. However, IRG1 was mainly expressed in macrophages, followed by neutrophils in liver cell subsets, whether and how macrophage-derived IRG1 and itaconate are involved in cancer progression remains unclear. Here, we demonstrated that macrophage-derived IRG1 is vital for HCC progression by antibody neutralization of macrophage and further revealed that macrophage-derived itaconate promotes HCC by inducing CD8+ T-cell exhaustion at the epigenetic level. Revitalization of exhausted T cells can reinvigorate immunity, suggesting that blocking the production of itaconate can reactivate T cells. This was demonstrated in a model in which ibuprofen suppresses HCC by alleviating itaconate synthesis. Moreover, itaconate can be degraded by

mammalian CLYBL (a ubiquitously expressed mitochondrial enzyme, conserved across all vertebrates) into pyruvate and acetyl-CoA[51] or by *Yersinia pestis* and *Pseudomonas aeruginosa*, representative bacteria with three genes for itaconate degradation[52]. Therefore, T-cell function can be boosted not only by inhibiting the synthesis of itaconate but also by degradation of endogenous itaconate, it may be possible to find ways to activate T cells from this perspective.

Inflammation is considered to be a local protective response of the body against injury or infection. However, in recent years, it has been established that long-term or excessive inflammation is involved in various stages of tumor. The interaction of cancer cells with surrounding immune cells forms an inflammatory tumor microenvironment, which drives tumorigenesis, growth, development, and transformation[53]. Chronic inflammation has a direct causal relationship and is one of the most important pathological features of liver cancer, 90% of primary HCC is transformed from chronic hepatitis. Therefore, given the link between inflammation and cancer, anti-inflammatory drugs have potential antitumor effects[54–56]. IRG1/itaconate is rapidly induced by activated macrophages, which are important components in the regulation of tissue inflammation. By screening commonly used clinical anti-inflammatory drugs, we found that ibuprofen has an obvious inhibitory effect on *Irg1* mRNA, protein, and itaconate production. Ibuprofen is a U.S. Food and Drug Administration (FDA)-approved nonsteroidal anti-inflammatory drug (NSAID) that has been used worldwide by consumers for decades to safely treat headaches, muscle aches, backaches, arthritis and other joint pain, but its role in tumors has not been realized or investigated. Three hundred milligrams up to 1200 mg of ibuprofen per day for adults weighing 60 kg (5–20 mg/kg) was used to treat the above conditions[57]. Here, we showed that 20 mg/kg ibuprofen twice weekly (approximately equal to 0.63 mg/kg daily in adults) could suppress HCC progression. Further in vivo experiments showed that the suppression of HCC progression by ibuprofen was reversed by 4-OI supplementation. It is worth noting that 20 mg/kg ibuprofen combined with low-dose anti-PD-1 antibody has a significant antitumor effect, demonstrating the potential of ibuprofen in clinical immunotherapy trials. These results indicate that low-dose ibuprofen, much lower than the dose of its clinical anti-inflammatory effect, has a potential therapeutic effect on tumors, greatly expanding its clinical application, especially in immunotherapy. However, how ibuprofen inhibits IRG1 expression is still not well understood. It is possible that ibuprofen may be involved in the inhibition of IRG1/itaconate by regulating the polarization of macrophages at the early stage.

In conclusion, this study identified a novel mechanism of macrophage-induced T-cell exhaustion by itaconate-mediated epigenetic remodeling in the TME, suggesting that targeting endogenous itaconate production may be an effective strategy to improve the antitumor efficacy of T cells. Moreover, clinical HCC patients with high macrophage expression had poor outcomes. We also found that ibuprofen suppressed HCC progression by reducing IRG1 expression and itaconate production. Importantly, ibuprofen combined with an anti-PD-1 antibody had a significant tumor suppressive effect, thus providing a potential therapeutic strategy for clinical HCC treatment.

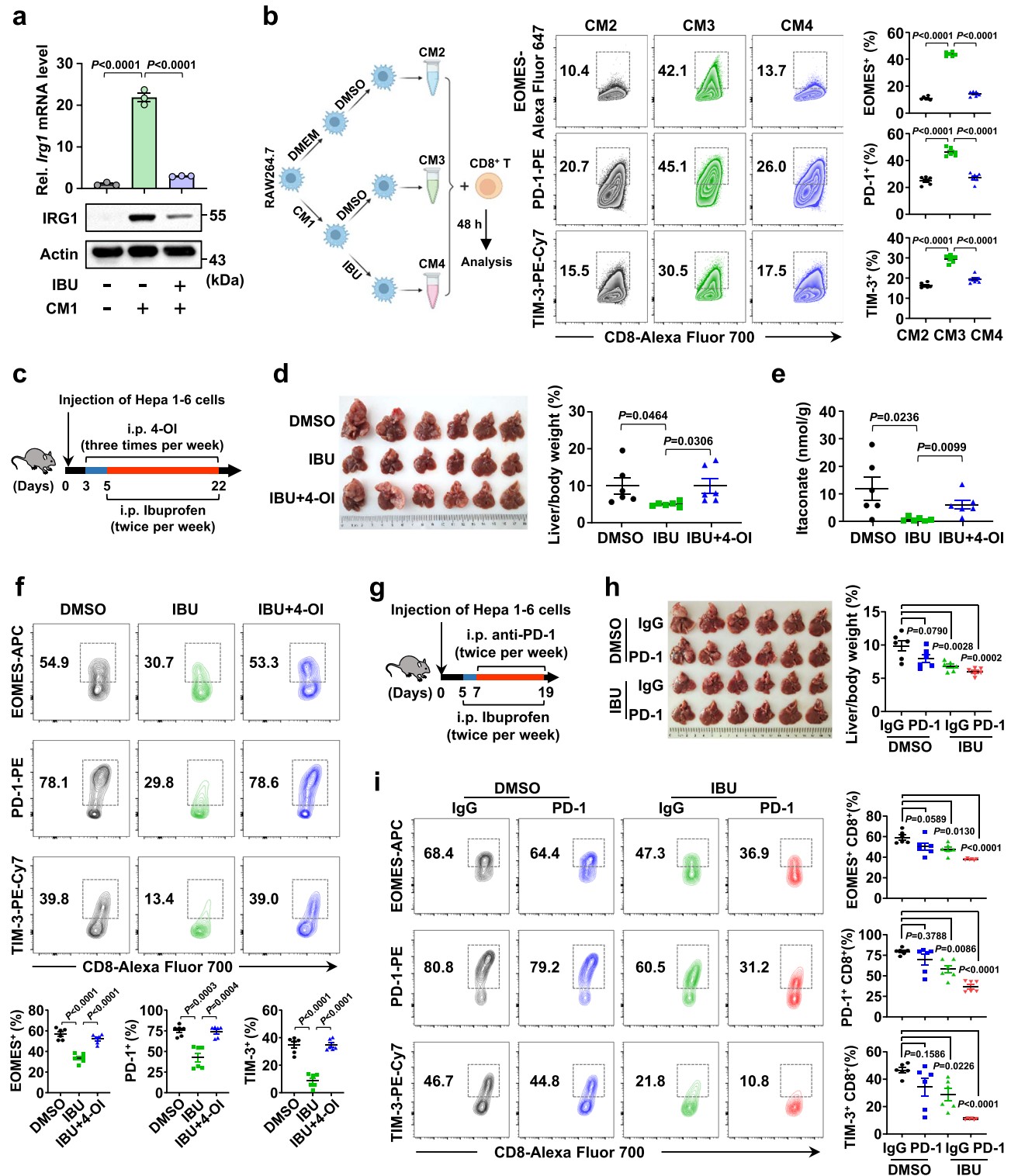

## Methods

### Mice

All animal experiments and procedures were performed in compliance with ethical regulations and the approval of the South China University of Technology (2023016). Mice were bred and housed at the animal facilities in ventilated cages with at most 6 animals per cage under specific-pathogen-free conditions. Mice feed was purchased from Beijing Keao Xieli Feed Co., Ltd (# 1016706476803973120). These mice were maintained in a 12 h light/12 h dark cycle, and the housing temperature and humidity were maintained at 24 °C and 50%, respectively.

Six- to eight-week-old male mice were used for all animal experiments unless otherwise stated. All mice were euthanized by $CO_2$ inhalation. Mice were monitored daily and euthanized based on defined criteria, including weight loss of 20% of pre-experimental body weight, anxiety, or other signs of distress. Orthotopic liver cancer was not grossly measurable and thus, tumor measurements were not routinely taken.

$Irg1^{-/-}$ mice (C57BL/6NJ-$Acod1^{em1(IMPC)}$/J, stock #029340) were purchased from the Jackson Laboratory. $Rag1^{-/-}$ mice (strain No. T004753) were purchased from GemPharmatech (Nanjing, China).

**Fig. 5 | Ibuprofen inhibits HCC by blocking IRG1/itaconate-regulated immune evasion. a** Analysis of *Irg1* mRNA and protein levels in RAW264.7 cells treated with ibuprofen after Hepa 1-6 supernatant (CM1) stimulation. **b** Schematic diagram of CD8+ T cells cocultured with different RAW264.7 cell-derived supernatants (left panel). RAW264.7 cells were treated with Hepa 1-6 cell supernatant (conditional medium 1, CM1), followed by treatment with or without ibuprofen, which was then replaced with fresh medium (the collected supernatants were denoted as CM3 or CM4). RAW264.7 cell-derived untreated supernatant was denoted as CM2. Finally, CD8+ T cells were cultured with CM2, CM3, or CM4 and analyzed by flow cytometry. Representative flow cytometry data and summary plot showing the percentages of EOMES+, PD-1+, and TIM-3+ cells among CD8+ T cells that were cocultured with RAW264.7 cells previously stimulated with Hepa 1-6 cell supernatant (right panel). **c** Schematic diagram of WT mice treated with ibuprofen or ibuprofen combined with 4-OI. **d** The liver images are shown, and the ratio of liver/body weight was determined. *n* = 6 mice per group. **e** UPLC–MS/MS analysis of the abundance of itaconate in the liver tissues of (**c**). **f** Representative flow cytometry data and summary plot showing the percentages of EOMES+, PD-1+, and TIM-3+ cells among CD8+ T cells in (**c**). *n* = 6 mice per group. **g** Schematic diagram of WT mice treated with ibuprofen or anti-PD-1 antibody. **h** Tumor images are shown (left panel), and the liver/body weight were measured (right panel). *n* = 6 mice per group. **i** Representative flow cytometry data and summary plot showing the percentages of EOMES+, PD-1+, and TIM-3+ cells among CD8+ T cells in (**g**). *n* = 6 mice per group. All data represent mean ± SEM. Statistical significance was determined by unpaired two-tailed Student's *t*-test (**d**, **e**) and one-way ANOVA (**a**, **b**, **f**, **h**, **i**). Data are representative of three independent experiments with similar results (**a**, **b**). Source data are provided as a Source Data file.

*Rag*−/− mice were crossed with *Irg1*−/− mice to generate *Irg1*+/+ *Rag*−/− mice (RWT) and *Irg1*−/− *Rag*−/− mice (RKO).

## Cell lines

Hepa 1-6 cells (ATCC) were cultured in DMEM with 10% FBS (Gibco) and 1% Pen/Strep. Jurkat cells (clone E6-1; ATCC) and RAW264.7 cells were cultured in RPMI-1640 (Gibco) with 10% FBS and 1% Pen/Strep. Mouse bone marrow-derived macrophages (BMDMs) were induced in complete DMEM supplemented with 20% L929 cell supernatant for 7 days. CD8+ T cells were cultured in RPMI-1640 (Gibco) with 10% FBS, 20 ng/ml IL-2 (PeproTech, 212-12), and 1% Pen/Strep after activation. All cell lines were tested for mycoplasma contamination and no cell lines were contaminated. Graphical abstract is provided in Supplementary Fig. 6.

## Mouse liver cancer models

**Hepa 1-6 cell-induced orthotopic liver cancer model.** Six- to eight-week-old C57BL/6 mice were injected with Hepa 1-6 cells ($1 \times 10^5$) through the hepatic portal vein to induce orthotopic liver cancer. (1) For neutralization experiments, mice were i.p. injected with anti-F4/80 antibody, anti-CD8α antibody (200 μg per mouse), or control immunoglobulin (rat anti-IgG2b antibody) two times per week from Day 3 after injection of Hepa 1-6 cells. (2) For 4-OI injection experiments, mice were i.p. injected with 4-OI (25 mg/kg) or DMSO three times per week from Day 3 after injection of Hepa 1-6 cells. (3) For the ibuprofen treatment experiment, mice were i.p. injected with ibuprofen (20 mg/kg) twice per week and 4-OI (25 mg/kg) three times per week from Day 3 after injection of Hepa 1-6 cells.

**Hepa 1-6 cell-induced subcutaneous tumor model.** Six- to eight-week-old C57BL/6 mice were injected subcutaneously with Hepa 1-6 cells ($5 \times 10^6$). (1) For anti-PD-1 antibody treatment experiments, mice were i.p. injected with anti-PD-1 antibody (20, 40, 60, and 100 μg per mouse) and control immunoglobulin (rat IgG2a) twice per week from Day 7 post-tumor implantation. (2) For the ibuprofen treatment experiment, mice were i.p. injected with ibuprofen (20, 40, and 80 mg/kg) twice per week from Day 5 post-tumor implantation. (3) For the ibuprofen and anti-PD-1 antibody combined treatment experiment, mice were i.p. injected with ibuprofen (20 mg/kg) and anti-PD-1 antibody (20 μg per mouse) twice per week from Day 5 post-tumor implantation.

**YAP^5SA-induced orthotopic liver cancer model.** Four-week-old C57BL/6 mice were hydrodynamically injected with *YAP^5SA*&*PB* (*piggyBac*) plasmids diluted in sterile Ringer's solution in a volume equal to 10% body weight. Approximately 90–100 days later, liver cancer was induced in mice in situ, and the mice were then euthanized for subsequent analyses. For the 4-OI treatment experiment, mice were i.p. injected with 4-OI (25 mg/kg) or DMSO three times per week from week 7 after hydrodynamic injection of *YAP^5SA*/*PB* plasmids. Approximately 80 days later, liver cancer was induced in mice in situ, and the mice were then euthanized for subsequent analyses.

**DEN-induced orthotopic liver cancer model.** Two-week-old C57BL/6 mice were i.p. injected with DEN (25 mg/kg) and 4 weeks later were injected with CCl$_4$ (1:4, v/v in olive oil) at a dose of 2 ml/kg body weight twice per week for 14 weeks. Liver cancer was induced in mice in situ, and then they were euthanized for subsequent analyses.

## Adoptive cell transfer

CD8+ T cells and CD4+ T cells isolated by microbeads (Miltenyi, negative) from wild-type mice were resuspended in PBS and i.v. injected into 6- to 8-week-old *Irg1*+/+ *Rag*−/− and *Irg1*−/− *Rag*−/− mice. Each mouse was inoculated with $1 \times 10^6$ CD8+ T cells or $2 \times 10^6$ CD4+ T cells. Mice were i.h. injected into the right flank with Hepa 1-6 cells ($2 \times 10^6$). (1) CD8+ T cells and CD4+ T cells were transferred to *Irg1*+/+ *Rag*−/− and *Irg1*−/− *Rag*−/− mice every 3 days from Day 12 post-tumor implantation. After 6 days, the mice were euthanized. (2) CD8+ T cells were adoptively transferred to *Irg1*+/+ *Rag*−/− mice and *Irg1*−/− *Rag*−/− mice 2 days before tumor implantation. After 19 days, the mice were euthanized.

## Coculture experiments

CD8+ T cells isolated from the spleen by microbeads (Miltenyi, negative) were stimulated and cultured with plate-bound anti-CD3 (2 μg/ml) and soluble anti-CD28 (1 μg/ml) antibodies under IL-2 (20 ng/ml) treatment for 3 days. CD8+ T cell and Hepa 1-6 cell coculture: CD8+ T cells were further treated with DMSO or 125 μM 4-OI for the next 2 days. After removing the supernatant, CD8+ T cells were cocultured with Hepa 1-6 cells at a ratio of 4:1 in a standard 10% RPMI-1640 T-cell medium containing 20 ng/ml IL-2 for 2 days. Hepa 1-6 cells treated with 4-OI (125 μM) for several days were used as a negative control to exclude the effect of 4-OI on Hepa 1-6 cells. CD8+ T cell and BMDM coculture: BMDMs from WT and KO mice were stimulated with 100 ng/ml LPS for 12 h before coculture with CD8+ T cells. CD8+ T cells were cocultured with BMDMs at a ratio of 1:1 in standard 10% 1640 T-cell medium containing 20 ng/ml IL-2 for 2 days. CD8+ T cell and hepatic macrophage populations coculture: F4/80+ cells (CD11b+ F4/80+), Kupffer cells (CD11b+ F4/80^high), and monocyte-derived macrophages (MoMs, CD11b^high F4/80+) were sorted from livers with Hepa 1-6 cell-induced tumor. Next, we cultured the sorted cells ($2.5 \times 10^5$/ml) separately in a transwell chamber. Then, CD8+ T cells from WT mice were cocultured with sorted cells in a 1:1 proportion for 48 h. CD8+ T cells cultured with RAW264.7 cell supernatant: RAW264.7 cells were stimulated with or without Hepa 1-6 cell supernatant for 24 h, treated with or without ibuprofen for 24 h, and then the medium was replaced with fresh medium for another 24 h. Finally, CD8+ T cells were cultured with RAW264.7 cell supernatant containing 20 ng/ml IL-2 for 48 h.

## Repetitive stimulation of CD8+ T cells

CD8+ T cells were cultured for 9 days in DMEM (10% FBS, 50 mM mercaptoethanol, 1 mM sodium pyruvate) as previously described. In detail, CD8+ T cells were stimulated with plate-bound anti-CD3/CD28 (1 μg/ml) and IL-2 (10 ng/ml) in the presence or absence of 4-OI (125 μM) for 3 days. The cells were then rested for 3 days in the

presence of only 10 ng/ml IL-2, followed by restimulation with plate-bound anti-CD3/CD28 and IL-2 in the presence or absence of 4-OI for another 3 days.

## qRT–PCR

Total RNA was extracted by TRIzol reagent (Life Technologies) and reverse transcribed with a cDNA synthesis kit (Vazyme). PCR was performed using SYBR Green master mix (Vazyme) and detected on a LightCycler 96 (Roche). The sequences of the primers used are shown in Supplementary Table 1. All samples were normalized to *18S* or *Actin* mRNA. (1) Detection of *Irg1* mRNA levels from mouse liver cell subsets: A single-cell suspension was obtained by hepatic portal vein perfusion of collagenase IV for 10 min. Immune cells were purified by density gradient centrifugation using Percoll. Next, CD8+ T cells, macrophages, NK cells, CD4+ T cells, monocytes, neutrophils, and B cells in the mouse liver were sorted by BD FACSAria SORP. Hepatic cells, stellate cells, and sinusoidal endothelial cells were isolated by digestion and independent density gradient centrifugation. The *Irg1* mRNA levels were normalized to the expression of mouse *Actin* mRNA. (2) Detection of transcription factors from mouse liver CD8+ T cells: Mouse liver tissues were digested in the presence of collagenase IV for 25 min prior to density gradient centrifugation using Percoll. Single-cell suspensions were incubated with microbeads (Miltenyi, positive) to enrich CD8+ T cells according to the manufacturer's protocol. Samples were normalized to *18S*. (3) Detection of transcription factors from Jurkat cells: Jurkat cells were treated with PHA (150 ng/ml) and 4-OI (125 μM) for 2 days. Total RNA was then collected to synthesize cDNA, followed by qRT-PCR for the detection of transcription factors. Samples were normalized to *18S*. (4) Detection of *Eomes*, *Pdcd1* and *Havcr2* mRNA levels: Jurkat cells were treated with PHA (150 ng/ml) and DMS or DMM (5 mM) for 12 h. CD8+ T cells were treated with DMS or DMM (5 mM) for 24 h. Total RNA was then collected to synthesize cDNA, followed by qRT-PCR for detection of transcription factors. Samples were normalized to *18S*.

## Chromatin immunoprecipitation quantitative real-time PCR (ChIP–PCR)

(1) CD8+ T cells were stimulated and amplified with plate-bound anti-CD3 (2 μg/ml) and soluble anti-CD28 (1 μg/ml) antibodies under IL-2 (20 ng/ml) treatment. After 3 days, cells were transferred to new wells and cultured in standard 10% 1640 T-cell medium in the presence of IL-2 for 12 h, followed by 4-OI (500 μM) treatment for 4 h. (2) Jurkat cells (1 × 10^6/ml) were stimulated with 1 μg/ml PHA for 2 h, and then the medium was replaced with fresh medium and the cells were treated with 4-OI (500 μM) for 4 h. (3) BMDMs isolated from WT mice were stimulated with 100 ng/ml LPS for 12 h, followed by ibuprofen treatment for 24 h. Next, cells were cross-linked in 37% formaldehyde and then terminated with 2.5 M glycine. Cells were then lysed, and the genome was sonicated into 200–1000 bp. The DNA cross-linking complex was immunoprecipitated with IgG antibody, H3K4me3 antibody or p50 antibody and subsequently de-cross-linked to enrich DNA, followed by qRT-PCR detection. Primer information is provided in Supplementary Table 2.

## Immunoblotting assay

**Immunoblotting of nuclear proteins.** For the 4-OI treatment assay, Jurkat cells were stimulated with 1 μg/ml PHA for 2 h, the medium was replaced with fresh medium, and the cells were treated with 4-OI (500 μM) for 4 h. For the DMS or DMM treatment assay, Jurkat cells were stimulated with 150 ng/ml PHA for 12 h in the presence of DMS or DMM (0, 2, 5 mM). For nuclear immunoblotting, cells were lysed with buffer A (150 mM NaCl, 1 mM KH_2PO_4, 1 mM MgCl_2, 1 mM PMSF, 0.2 mM DTT, and 0.3% Triton X-100, pH 6.4) supplemented with a protease inhibitor cocktail for 10 min to dissolve cell membranes. Then, extracted nuclear components were lysed with buffer B (50 mM

Tris-HCl, pH 7.5, 500 mM NaCl, 1 mM EDTA, 0.2% NP-40, 10 mM β-mercaptoethanol, and 10% glycerol) supplemented with protease inhibitor cocktail, followed by sonication with an Ultrasonic Cell Disruptor (Scientz). Protein levels of H3K4me1/2/3, H3K9me1/2/3, H3K27me3 and histone 3 were fractionated by 12% SDS–polyacrylamide gel electrophoresis (SDS–PAGE). Protein levels of EOMES were fractionated by 10% SDS–PAGE.

**Immunoblotting of whole-cell lysates.** BMDMs and RAW264.7 cells were stimulated with 100 ng/ml LPS for 12 h and then treated with the indicated drugs for 24 h. Cell lysates were prepared with lysis buffer (50 mM Tris-Cl, pH 8.0, 150 mM NaCl, 5 mM EDTA, 0.1% SDS, 1% NP-40, 10 mM NaF, 1 mM PMSF, 2 mM DTT, and 1 mM Na_3VO_4) supplemented with protease cocktails. Protein levels of IRG1 were fractionated by 10% SDS–PAGE. Signals were detected using Western ECL Substrate (Tannon).

## Hematoxylin-eosin staining

Mouse liver tissues were fixed with 10% neutral formalin for 24 h, embedded in paraffin, sectioned, and stained with hematoxylin and eosin. In brief, for deparaffinization, the slide was heated in an oven and placed in xylene to remove the wax. For hydration, the xylene was washed off, and the tissue was hydrated by a gradient concentration series of alcohol (100%, 90%, 80%, and 70%) and water. Nuclear staining was accomplished by staining in hematoxylin for 2–3 min. The slides were washed in running tap water until the sections were "blue". For the differentiation step, slides were dipped in 1% acid alcohol (1% HCl in 70% alcohol) for a few seconds to remove excess dye. Then, they were washed in running tap water until the sections were "blue". Counterstaining was performed by staining in 1% Eosin Y for 1–2 min, followed by washing in tap water for 3–5 min. Samples were dehydrated in increasing concentrations of alcohol and then cleared by cleaning the slides in xylene twice. The sections were mounted in neutral resins. Images were captured using a Leica microscope (Aperio CS2). Whole-slide images were processed using HALO (v3.3) software.

## Flow cytometry

C57BL/6 mouse liver tissues were digested in the presence of collagenase IV prior to density gradient centrifugation using Percoll. Single-cell suspensions were stained with anti-CD16/32 antibodies against surface molecules. For cell membrane protein staining, cells were stained at 4 °C for 30 min. For intracellular cytokine staining, cells were incubated with a stimulation cocktail for 4 h prior to cell surface and cytokine staining. Then, cell staining was performed after fixation and permeabilization with antibodies against the murine samples. All data were collected on a BD Fortessa or Cytek NL-3000 and analyzed with FlowJo (10.8.1) software. Gating and sorting strategies were provided in Supplementary Fig. 7.

## UPLC–MS/MS analysis for succinate and itaconate

Detection of itaconate and succinate in BMDMs: WT mice and KO mice were stimulated with 100 ng/ml LPS for 12 h or Hepa 1-6 cell supernatant for 24 h. To detect intracellular and extracellular succinate and itaconate abundance, the same supernatant volume (6 ml per sample) and cell number (6 × 10^6 per sample) were collected. Detection of itaconate in CD8+ T cells: CD8+ T cells (1 × 10^6/ml) from WT mice were treated with MCT1 inhibitors (50 μM) for 30 min before itaconate (5 mM) treatment for 1 h (8 × 10^6 per sample). An equal amount of supernatant from each sample was used as a QC sample. UPLC separation was conducted on a UPLC BEH C18 column (100 mm × 2.1 mm, 1.7 μm). The mobile phase consisted of 0.1% formic acid (A) and 0.1% formic acid–methanol (B). The gradient program was as follows: 0–1 min (5% B), 1–1.5 min (5–10% B), 1.5–8.5 min (10–100% B), 8.5–10.5 min (100% B), 10.5–11 min (100–5% B), 11–12 min (5% B); flow rate, 0.3 mL/min. The column oven temperature was maintained at

40 °C. Itaconate and succinate in cells, cell supernatant and mouse liver tissues were measured by ultra-performance liquid chromatography coupled to a tandem-mass spectrometry (UPLC–MS/MS, ACQUITY UPLC-Xevo TQ-S, Waters Corp., Milford, MA, USA) system (Provided by Metabo-Profile R& D Laboratory, Shanghai, China). Masslynx (v4.1, Waters, Milford, MA, USA) and iMAP (v1.0, Metabo-Profile, Shanghai, China) were used to analyze metabolic data.

### Detection of intracellular levels of succinate, α-KG, and fumarate

Intracellular levels of succinate, α-KG, and fumarate from 4-OI (500 μM)-treated CD8[+] T cells were determined according to the manufacturer's protocol. In brief, CD8[+] T cells were rapidly homogenized on ice and centrifuged at maximum speed for 5 min at 4 °C. The supernatant was then collected, added to the reaction mixture and incubated at 37 °C for the indicated time. The output was measured on a microplate reader (Biotek-800TS).

### Overall survival analysis of liver hepatocellular carcinoma (LIHC)

LIHC gene expression and survival data were downloaded from TCGA datasets. The samples were grouped based on the expression levels of *IRG1* and *CD68* by the median value. The overall survival curves were plotted to show differences in survival time, and log-rank *p*-values reported by the Cox regression models implemented in the R package survival were used to determine the statistical significance. The samples were normalized and filtered by the TCGA analyze-Normalization and TCGA analyze-Filtering functions in R (version 4.0.5). The overall survival curves of the four groups of patients were analyzed and drawn with the survival R package (version 3.2-13), survminer R package (version 0.4.9), and survMisc R package (version 0.5.5). TCGA LIHC survival information is provided in Source Data file.

### t-distributed stochastic neighbor embedding (t-SNE) analysis

Single liver cells were isolated, stained, and tested with Fortessa flow cytometry. Flow cytometry data were analyzed with FlowJo software and R language (Version 4.0.5). CD45-positive cells were exported as an expression matrix based on the channel values of all samples in FlowJo software. Data from 5000 randomly selected cells were merged into one matrix, normalized by channel, and analyzed with the t-SNE algorithm by the Seurat package (version 4.0.2). The parameters used for t-SNE dimensionality reduction analysis were CD3, CD8α, CD4, CD11b, CD11c, CD19, F4/80, Ly-6G/Ly-6C (Gr-1), MHC-II, NK1.1, and PD-1. Individual analysis information is provided in Supplementary Fig. 8.

### Ethical statement

We have complied with all relevant ethical regulations for animal testing and research. The research protocol was approved by the Animal Experiment Ethics Committee of the South China University of Technology. The mice with orthotopic tumor, authorized by the Committees on Animal Research and Ethics, consistently follow the humane endpoint. The subcutaneous tumor maximum diameter was 20 mm and authorized by the Committees on Animal Research and Ethics and was not exceeded at any time during the experiments.

### Statistics and reproducibility

Unpaired two-tailed Student's *t*-test and one- and two-way analysis of variance (ANOVA) were used to calculate *P*-values by GraphPad Prism 9 unless otherwise indicated in the figure legends. The Tukey method was used to adjust multiple comparisons. All data represent mean ± SEM. $P < 0.05$ was considered significant. ns, no significant difference. Kaplan-Meier curves were used to depict survival function from lifetime data for human patients using the log-rank test. Mice were randomly grouped before different treatments. In vitro studies, cells or

conditions were assigned randomly to each experimental group. Before performing H&E, the researchers took the same spot in the mouse liver to rule out potential bias caused by the subjective selection of tumor sites. Investigators were blinded for H&E analysis. In other experiments, samples were analyzed in a blinded manner without subjective estimation. We defined each sample in different groups performing three independent biological experiments. For animal studies, we used at least three C57BL/6 mice for different groups.

### Reporting summary

Further information on research design is available in the Nature Portfolio Reporting Summary linked to this article.

## Data availability

Experimental resources of antibodies, chemicals, reagents and critical commercial assays were provided in Supplementary Table 3. TCGA LIHC gene expression and survival data were downloaded from https://www.cancer.gov/aboutt-nci/organization/ccg/research/structural-genomics/tcga. Plasmids generated in this study are available. Further information and requests for resources and reagents should be directed to the lead contact, P.G. (pgao2@ustc.edu.cn). Source data are provided with this paper.

## Code availability

This paper does not report the original code.

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

## Acknowledgements

This work is supported in part by National Key R&D Program of China (2018YFA0800300, 2022YFA1304504), National Natural Science Foundation of China (82130087, 82341013, 82192893, 81930083, 81821001, 82273221, 82203556), the Chinese Academy of Sciences

(XDB39000000), the Global Select Project (DJK-LX-2022001) of the Institute of Health and Medicine, Hefei Comprehensive National Science Center, the Fundamental Research Funds for the Central Universities (YD2070002008). The gray mouse cartoon in Figs. 1a, b, c, f, 2b, 3b, 5c, g, Supplementary Figs. 1d, 3d, 5f, i, k were created with Motifolio.com. The schematics in Figs. 2e, g, 3d, e, 4e, h, 5b, Supplementary Fig. 3e, g and Supplementary Fig. 6a were created with BioRender.com.

## Author contributions

P.G. conceived and supervised the study. X.G., L.S., P.G., and H.Z. designed the experiments. X.G., H.W., C.S., S.S., C.Z., L.C., K.Y., Z.L., Z.B., P.Z., M.Y. and Y.Y. performed and analyzed the experiments. J.D. provided constructive suggestions. L.S., X.G., and P.G. wrote the manuscript. All the authors read and approved the manuscript.

## Competing interests

The authors declare no competing interests.
