## [Peer Review File · Nature Communications]

Itaconate promotes hepatocellular carcinoma progression by epigenetic induction of CD8+ T-cell exhaustionEditorial Note: Parts of this Peer Review File have been redacted as indicated to remove third-party material where no permission to publish could be obtained.

REVIEWER COMMENTS

Reviewer #1 (Remarks to the Author):

The study by Gu, Suo, et al. demonstrates that itaconate, a metabolite produced by macrophages, promotes immune evasion in hepatocellular carcinoma (HCC). Given the low rate of response to HCC patients to immunotherapies, this is a very relevant question. By using IRG1-KO mice, which are unable to produce itaconate, the authors found a reduction in HCC in 3 separate mouse models. The effects were mediated by macrophages and rescued by administration of itaconate. Mechanistically, itaconate led to CD8 T cell exhaustion by promoting expression of EOMES. Finally, ibuprofen was shown to inhibit IRG1 and in turn, HCC, improving response to anti-PD1.

The study is novel, original, relevant, and rigorously conducted. There are only a few minor questions that remain unanswered:

- It would be important to show total levels of CD8 T cells in the livers of WT and KO mice harboring tumors. Are KO mice enriched in CD8 T cells?
- The combination of ibuprofen and aPD1 shows significant therapeutic effects. It would be important to repeat this experiment in an intrahepatic tumor model.
- It would be important to conduct at least one experiment (for example, experiment shown in Fig. 1A) in females.

Reviewed by [EDITORIAL NOTE: REVIEWER NAME REDACTED AT OWN REQUEST.]

Reviewer #2 (Remarks to the Author):

In this manuscript the authors investigated the role of macrophage-derived itaconate on CD8 T-cell exhaustion in HCC. The authors uncovered here, that macrophage-derived itaconate increases succinate levels in hepatic CD8 T-cells by SDH inhibition, which leads to

an upregulation of the exhaustion markers PD-1 and TIM-3 via Eomes. This represents a novel mechanism how the hepatic immune environment is suppressing the immune surveillance leading to HCC formation and progression. Moreover, they describe that ibuprofen treatment successfully suppresses itaconate formation and CD8 T-cell exhaustion and therefore synergizes with anti PD-1 therapy.

However, there are some points which need to be addressed:

- The authors showed an increase in Irg1 expression of macrophages compared to all other investigated cell populations (Figure 1D). In recent years, hepatic macrophage populations were characterized extensively which led to the distinction of several functionally different subsets (monocyte derived macrophages, resident Kupffer cells...). Are there differences between these hepatic macrophage populations in Irg1 expression and their roles in itaconate-mediated induction of T-cell exhaustion?
- The authors demonstrated an impressive treatment effect by combining ibuprofen and anti-PD1. Nevertheless, the mechanistic link between the COX-inhibitor ibuprofen and Irg1/itaconate is not clear. Could the authors demonstrate how the ibuprofen interferes with Irg1/itaconate and how they selected the group of drugs which they screened in Suppl Fig 5A?
- MDSCs have been recently reported to suppress CD8+ T cells and promote tumor growth via itaconate (Zhao et al., Nat Metabolism 2022). Is there a role of itaconate derived by hepatic MDSCs in hepatic CD8 T cell exhaustion too?
- Throughout the Figure 3-5, the effect of itaconate inducing T cell exhaustion has been well demonstrated. Nevertheless, all the experiments are done with the cell-permeable form of itaconate (4-OI). How do T cells normally internalize itaconate?
- Authors have demonstrated that accumulated succinate can induce EOMES expression via histone methylation which in turn induces T cell exhaustion. However, the mechanism of how accumulated succinate increase H3K4me3 is still not entirely clear.

Minor points:

- Which of the three HCC mouse models have been used for the Data shown in Figure 1D?
- Could authors check whether all abbreviations have been correctly introduced (e.g. SDH)?

Point-by-point response to the comments of the Reviewers

We appreciate the reviewers for their positive, constructive, and insightful comments and suggestions which have greatly assisted us in preparing the revised manuscript. In the past three months, we have performed additional new experiments and addressed all the concerns and comments raised by our reviewers. Here, we resubmit a substantially improved manuscript along with our point-by-point responses. For the convenience of our reviewers, we have appended in this file all the revised figures, which we labeled as **Figure R1** to **Figure R7**.

Reviewers Comments

Reviewer #1 (Remarks to the Author):

The study by Gu, Suo, et al. demonstrates that itaconate, a metabolite produced by macrophages, promotes immune evasion in hepatocellular carcinoma (HCC). Given the low rate of response to HCC patients to immunotherapies, this is a very relevant question. By using IRG1-KO mice, which are unable to produce itaconate, the authors found a reduction in HCC in 3 separate mouse models. The effects were mediated by macrophages and rescued by administration of itaconate. Mechanistically, itaconate led to CD8 T cell exhaustion by promoting expression of EOMES. Finally, ibuprofen was shown to inhibit IRG1 and in turn, HCC, improving response to anti-PD1.

The study is novel, original, relevant, and rigorously conducted. There are only a few minor questions that remain unanswered:

Response: We are grateful for the positive comments of our reviewer, which effectively summarized the major findings and implication of our study. We also appreciate his/her constructive comments and suggestions for this manuscript.

1. It would be important to show total levels of CD8 T cells in the livers of WT and KO mice harboring tumors. Are KO mice enriched in CD8 T cells?

Response: We appreciate this valuable suggestion. Accordingly, we analyzed the ratio of CD8⁺ T cells in the livers of wild type (WT) and *Irg1*^{-/-} (KO) mice harboring tumors induced by DEN/CCL₄ (**Figure R1A**) or Hepa 1-6 (**Figure R1B**). Our results showed

that there was no significant difference in the proportion of CD8⁺ T cells between WT and KO mice (**Figure R1A-B**). Therefore, these data reinforce our conclusion that itaconate promotes exhaustion of CD8⁺ T cells without affecting total levels of CD8⁺ T cells.

Figure R1. CD8⁺ T cells proportions of WT and KO mice in different liver tumor models. (A) Representative flow cytometry data and summary plot of the frequency showing the percentage of CD8⁺ T cells among T cells isolated from livers of wild type (WT) and *Irg1*^{-/-} (KO) mice with DEN/CCl₄-induced liver cancer. *n*=8 mice per group. (B) Representative flow cytometry data and summary plot of the frequency showing the percentage of CD8⁺ T cells among T cells isolated from livers of WT and KO mice with Hepa 1-6 cell-induced liver cancer. *n*=5 mice per group. ns, not significant; unpaired Student's *t* test. Data are the mean ± SEM.

2. The combination of ibuprofen and aPD1 shows significant therapeutic effects. It would be important to repeat this experiment in an intrahepatic tumor model.

Response: As suggested by the reviewer, we administered intrahepatic injection of Hepa1-6 cells into WT mice, followed by a combination treatment of low-dose anti-PD-1 antibody and low-dose ibuprofen (**Figure R2A**). Our *in vivo* combination experiment demonstrated a significant enhancement in the antitumor efficacy of the anti-PD-1 antibody when combined with ibuprofen (**Figure R2B-C**). These data are consistent with the results in original Fig. 5G-H (Supplementary Fig. 5k, l in the revised manuscript). We have included **Figures R2A-C** as **Fig. 5g-i** in the revised manuscript.

Figure R2. The combination of ibuprofen and anti-PD-1 antibody shows significant therapeutic effects in intrahepatic tumor model. (A) Schematic diagram of the Hepa 1-6 cell-induced HCC model. WT mice at 8 weeks were injected with 1×10^5 Hepa 1-6 cells, and 5 days later, they were treated with ibuprofen (20 mg/kg) or ibuprofen (20 mg/kg) combined with anti-PD-1 antibody (20 μ g per mouse). Approximately 3 weeks later, the mice were sacrificed. (B) Tumor images are shown (left panel), and the liver/body weight ratio was measured at the end of the experiment (right panel). $n=6$ male mice per group. ns, not significant, $*P<0.05$, $**P<0.01$; one-way ANOVA (Tukey's multiple comparisons test). Data are the mean \pm SEM. (C) Representative flow cytometry data and summary plot showing the percentages of EOMES⁺, PD-1⁺, and TIM-3⁺ cells among CD8⁺ T cells in **Figure R2B**. $n=6$ male mice per group. ns, not significant, $*P<0.05$, $**P<0.01$; one-way ANOVA (Tukey's multiple comparisons test). Data are the mean \pm SEM. We have included **Figures R2A-C** as **Fig. 5g-i** in the revised manuscript.

3. It would be important to conduct at least one experiment (for example, experiment shown in Fig. 1A) in females.

Response: We appreciate the reviewer's important suggestion. We thus administered intrahepatic injection of Hepa1-6 cells into female WT and KO mice. Three weeks later,

we found that knockout of IRG1 obviously alleviated the tumorigenesis in female mice (**Figure R3A**), which is consistent with the phenomenon we observed in male mice as shown in the original Fig. 1A. We then analyzed the exhaustion markers of CD8⁺ T cells in the livers of WT and KO mice, and the results showed that the numbers of tumor-infiltrating EOMES⁺ CD8⁺, PD-1⁺CD8⁺, and TIM3⁺CD8⁺ T cells in the livers of female KO mice was reduced (**Figure R3B**). These results suggested that IRG1 deficiency suppresses HCC progression is sex-independent. We have included **Figures R3A, R3B** as **Supplementary Fig. 1d, 4c** in the revised manuscript, respectively.

Figure R3. Loss of IRG1 also suppresses HCC progression in female mice. (A) Schematic diagram of the Hepa 1-6 cell-induced HCC model (left panel). WT and KO mice at 8 weeks were injected with 1×10^5 Hepa 1-6 cells through the hepatic portal vein. Three weeks later, the mice were sacrificed. Representative liver images are shown (middle panel), and the ratio of liver/body weight was determined (right panel). $n=5$ female mice per group. $*P < 0.05$; unpaired Student's t test. Data are the mean \pm SEM. (B) Representative flow cytometry data and summary plot showing the percentages of EOMES⁺, PD-1⁺, and TIM-3⁺ cells among CD8⁺ T cells in **Figure R3A**. $n=5$ female mice per group. $*P < 0.05$; unpaired Student's t test. Data are the mean \pm SEM. We have included **Figures R3A, R3B** as **Supplementary Fig. 1d, 4c** in the revised manuscript, respectively.

Reviewer #2 (Remarks to the Author):

In this manuscript the authors investigated the role of macrophage-derived itaconate on CD8 T-cell exhaustion in HCC. The authors uncovered here, that macrophage-derived itaconate increases succinate levels in hepatic CD8 T-cells by SDH inhibition, which

leads to an upregulation of the exhaustion markers PD-1 and TIM-3 via Eomes. This represents a novel mechanism how the hepatic immune environment is suppressing the immune surveillance leading to HCC formation and progression. Moreover, they describe that ibuprofen treatment successfully suppresses itaconate formation and CD8 T-cell exhaustion and therefore synergizes with anti PD-1 therapy. However, there are some points which need to be addressed:

Response: We would like to express our gratitude to the reviewer for the positive comments and summary of our discovery. Meanwhile, we thank the reviewer for his/her valuable advice, which has been helpful in strengthening this study.

1. The authors showed an increase in *Irg1* expression of macrophages compared to all other investigated cell populations (Fig. 1D). In recent years, hepatic macrophage populations were characterized extensively which led to the distinction of several functionally different subsets (monocyte derived macrophages, resident Kupffer cells...). Are there differences between these hepatic macrophage populations in *Irg1* expression and their roles in itaconate-mediated induction of T-cell exhaustion?

Response: We thank the reviewer for the very important point. Following this suggestion, we sorted F4/80⁺ cells (CD11b⁺ F4/80⁺), Kupffer cells (CD11b⁺ F4/80^{high}), and monocyte-derived macrophages (MoMs, CD11b^{high} F4/80⁺) from livers with Hepa 1-6 cell-induced tumor (**Figure R4A**). The qRT-PCR results showed no significant variation in the expression of *Irg1* mRNA levels among F4/80⁺ cells, Kupffer cells, and MoMs (**Figure R4B**). To investigate the role of *Irg1* and itaconate of these cells in T-cell exhaustion, CD8⁺ T cells from WT mice were cocultured with sorted cells in a 1:1 proportion for 48 h (**Figure R4A**). By analyzing the percentages of EOMES⁺, PD-1⁺, and TIM-3⁺ cells among CD8⁺ T cells, we observed no significant difference between Kupffer cells and MoMs in itaconate-mediated CD8⁺ T-cell exhaustion compared to F4/80⁺ cells (**Figure R4C**). We have included **Figures R4A-C** as **Supplementary Fig. 3g-i** in the revised manuscript, respectively.

Figure R4. *Irg1* expression in hepatic macrophage populations and their roles in itaconate-mediated induction of T-cell exhaustion. (A) Schematic diagram of population sorting of hepatic macrophage from tumor-bearing mice and co-culture of the sorting cells with CD8⁺ T cells. F4/80⁺ cell (CD11b⁺ F4/80⁺), Kupffer cell (CD11b⁺ F4/80^{high}) and MoM (monocyte-derived macrophage, CD11b^{high} F4/80⁺) were sorted by flow cytometry. (B) Analysis of *Irg1* mRNA levels in hepatic macrophage populations sorted from Hepa 1-6 cell-induced mouse liver. *n*=5 male mice per group. Data are the mean ± SEM. (C) Representative flow cytometry data and summary plot showing the percentages of EOMES⁺, PD-1⁺, and TIM-3⁺ cells among CD8⁺ T cells. *n*=3 samples per group, cells from 5 mice were mixed as one sample. Data are the mean ± SEM. We have included **Figures R4A-C** as **Supplementary Fig. 3g-i** in the revised manuscript, respectively.

2. The authors demonstrated an impressive treatment effect by combining ibuprofen and anti-PD1. Nevertheless, the mechanistic link between the COX-inhibitor ibuprofen

and *Irg1*/itaconate is not clear. Could the authors demonstrate how the ibuprofen interferes with *Irg1*/itaconate and how they selected the group of drugs which they screened in Suppl Fig 5A?

Response: We appreciate the reviewer's comments. Previous studies have reported that ibuprofen showed inhibitory effect on nuclear transcription factor-kappa B (NF- κ B) (**Figure R5A**) (Palayoor ST et al., 1999; Zhang NY et al., 2022), and knockout of p50 (NF- κ B subunit) reduced the expression of *Irg1* in mouse bone marrow-derived macrophages (BMDMs) (**Figure R5B**) (Bomfim CCB et al., 2022). Through further analysis of the potential transcriptional factors binding to *Irg1* promoter (<https://jaspar.genereg.net/>), we found that NF- κ B binds to GGGGCTTTTC (chr14: 103284082-103284092), TGGAAATTCC (chr14: 103284362-103284372), and other sequences of *Irg1* promoter regions (**Figure R5C**). To investigate the relationship between p50 and *Irg1*, we treated BMDMs with ibuprofen (IBU) and found attenuated binding activity of p50 to the promoter of the *Irg1* gene by ChIP-qPCR (**Figure R5D**). We thus conclude that the ibuprofen interferes with *Irg1*/itaconate, at least partially, through NF- κ B. We have included **Figure R5D** as **Supplementary Fig. 5d** in the revised manuscript.

As for the second question, considering that many studies have shown that IRG1/itaconate is closely related to inflammation (Peace CG et al. 2022; Mills EL, et al. 2018) and that the role of IRG1/itaconate in liver cancer is the focus of this study, the clinical analgesic-antipyretic and anti-hepatitis drugs are important concerns during our screening for targeting IRG1 drugs.

[FIGURES REDACTED]

C

Name	Score	Relative score	Sequence ID	Start	End	Strand	Predicted sequence
MA0105.1.NFKB1	10.367545	0.8866313490531389	NC_000080.7:103282448-103294447	1916	1925	-	TGGAATTTC
MA0105.1.NFKB1	9.836401	0.8737684972024627	NC_000080.7:103282448-103294447	1635	1644	+	GGGGCTTTTC
MA0105.1.NFKB1	8.626999	0.8444801014462662	NC_000080.7:103282448-103294447	1915	1924	+	TGGAATTTC
MA0105.1.NFKB1	7.7525744	0.8233039434893568	NC_000080.7:103282448-103294447	2487	2496	-	GGCAAGTTCC
MA0105.1.NFKB1	7.1127434	0.8078089929477146	NC_000080.7:103282448-103294447	1693	1702	-	TGGGCATCCC

Figure R5. Ibuprofen inhibits *Irg1* level via NF-κB (p50). (A) Immunoblotting quantitative analyses of total p65 and p-p65 in the hippocampus of the control, CIA, and CIA+Ibu groups (Zhang NY et al., 2022; Figure 4L). Rats with collagen-induced arthritis were recorded as "CIA". n=3, 3–6 animals per group. *P<0.05; unpaired Student's t test. Data are the mean ± SEM. (B) *Irg1* mRNA expression was assessed in C57BL/6 (BL/6) and NFκB^{-/-} (p50^{-/-}) BMDMs after 6 h post-injection (pi) of Mycobacterium tuberculosis (Bomfim CCB et al., 2022; Figure 2B). (C) List of binding sequences predicted by NF-κB in the *Irg1* promoter regions. (D) ChIP experiments were performed in BMDMs isolated from WT mice treated with DMSO or ibuprofen for 24 h using IgG or p50 antibodies. BMDMs were stimulated with 100 ng/ml lipopolysaccharide (LPS) for 12 h before ibuprofen treatment. The occupancy of potential binding site (GGGGCTTTTC) in the *Irg1* gene by NF-κB (p50) was determined by qRT-PCR. The experiments were repeated three times independently with similar results. **P<0.01; unpaired Student's t test. Data are the mean ± SEM. We have included **Figure R5D** as **Supplementary Fig. 5d** in the revised manuscript, respectively.

3. MDSCs have been recently reported to suppress CD8+ T cells and promote tumor growth via itaconate (Zhao et al., Nat Metabolism 2022). Is there a role of itaconate derived by hepatic MDSCs in hepatic CD8 T cell exhaustion too?

Response: Following the reviewer's suggestion, we sorted MDSCs from WT and KO mouse liver and cultured these MDSCs (2.5×10⁵/ml) in 12-well plates for 5 days. To exclude the intrinsic influence caused by tumor size between WT and KO mice, we

induced MDSCs with Hepa 1-6 cell supernatant for 24 h. Then, hepatic MDSCs were cocultured with CD8⁺ T cells from WT mice in a 1:1 proportion for 48 h (**Figure R6A**). Analysis of the percentages of EOMES⁺, PD-1⁺, and TIM-3⁺ cells among CD8⁺ T cells showed that hepatic MDSCs-derived itaconate also induces CD8⁺ T-cell exhaustion (**Figure R6B**), although its effect was lower than that of macrophage-derived itaconate as shown in the original Fig. 3D.

Figure R6. Hepatic MDSCs-derived itaconate promoted CD8⁺ T-cell exhaustion. (A) Schematic diagram of hepatic MDSCs cocultured with CD8⁺ T cells. MDSCs were sorted from Hepa 1-6 cell-induced WT and KO mouse liver cancer by Myeloid-Derived Suppressor Cell Isolation Kit (Miltenyi, no. 130-094-538). (B) Representative flow cytometry data and summary plot showing the percentages of EOMES⁺, PD-1⁺, and TIM-3⁺ cells among CD8⁺ T cells. *n*=7 samples per group, cells from 2 mice were mixed as one sample. Data are the mean ± SEM.

4. Throughout the Figure 3-5, the effect of itaconate inducing T cell exhaustion has been well demonstrated. Nevertheless, all the experiments are done with the cell-permeable form of itaconate (4-OI). How do T cells normally internalize itaconate?

Response: We appreciate the insightful comments of our reviewer. There is currently no available report on the mechanism by which T cells internalize itaconate. However, given the similar chemical structure of itaconate and succinate (**Figure R7A**), we speculate that itaconate may shuttle into the interior of the cells through the same membrane carrier monocarboxylate transporter 1 (MCT1) utilized by succinate. Three compounds, namely 7ACC2, AZD3965, and BAY-8002, have been reported as inhibitors of MCT1 to impede the entry of succinate into cells (Wang N et al., 2021). Meanwhile, succinate can be absorbed by T cells via MCT1 (Gudgeon N et al., 2022). Furthermore, our simulation shows that both itaconate and succinate bind to the substrate pocket of MCT1, which indicates that MCT1 can transport itaconate into cells (**Figure R7A**). Those lines of evidence suggest that T cells may internalize itaconate via MCT1. To test this hypothesis, CD8⁺ T cells ($1 \times 10^6/\text{ml}$) from WT mice were treated with MCT1 inhibitors (50 μM) for 30 min before itaconate (5 mM) treatment for 1 h, then itaconate levels was detected by UPLC-MS/MS. The results showed that intracellular itaconate is indeed partially reduced when treating CD8⁺ T with MCT1 inhibitors. Taken together, these results suggested that MCT1 is a potential transporter mediates the internalization of itaconate to CD8⁺ T cells (**Figure R7B**). We have included **Figures R7A, B** as **Supplementary Fig. 4f, g** in the revised manuscript, respectively.

A

B

Figure R7. CD8⁺ T cell partly internalized itaconate via MCT1. (A) Docking model of MCT1 (pdb:6lyy) bound to multiple ligands (AZD3965, succinate, itaconate) with visualization of transporter tunnel. AZD3965 (blue and yellow); Succinate (pink); Itaconate (cyan); MCT1 (white and grey). (B) UPLC–MS/MS analysis of the abundance of itaconate in the CD8⁺ T cells. CD8⁺ T cells isolated from WT mouse spleens were stimulated with plate-bound anti-CD3/CD28 and IL-2, recorded as “Active”. Unstimulated CD8⁺ T cells were recorded as “Naïve”. Activated CD8⁺ T cells were treated with MCT1 inhibitors (50 μ M) for 30 min before itaconate (5 mM) treatment for 1 h, followed by itaconate detection. * P <0.05; one-way ANOVA (Tukey’s multiple comparisons test). Data are the mean \pm SEM. We have included **Figures R7A, B** as **Supplementary Fig. 4f, g** in the revised manuscript, respectively.

5. Authors have demonstrated that accumulated succinate can induce EOMES expression via histone methylation which in turn induces T cell exhaustion. However, the mechanism of how accumulated succinate increase H3K4me3 is still not entirely clear.

Response: We appreciate the important question. It’s known that the absence of succinate dehydrogenase (SDH) activity in tumors leads to elevated cellular succinate levels which subsequently hinders the function of Jumonji-C domain-containing histone demethylases (JHDMs) and results in epigenetic reprogramming including H3K4me3 (Cervera AM et al., 2009; Letouzé E et al., 2013; Xiao M et al., 2012; Hoekstra AS et al., 2015; Janzer A et al., 2012). Furthermore, our recent work showed that the accumulation of succinate leads to an increase in H3K4me3 levels (Li ST et al., 2020). Our result in T cells is consistent with the above findings and, potentially, the same mechanism might apply to our observations in this manuscript.

Minor points:

1. Which of the three HCC mouse models have been used for the Data shown in Figure 1D?

Response: As we mentioned in the manuscript, Fig. 1d shows *Irg1* mRNA levels of different cells within the mouse liver under steady-state conditions without any treatment or stimulation.

2. Could authors check whether all abbreviations have been correctly introduced (e.g. SDH)?

Response: We thank the reviewer for pointing this out. We revised “SDH” to “succinate dehydrogenase (SDH)” when it first appeared in the **Results** section. “Eomesodermin

(EOMES)” in the **Abstract** section, and “tumor-infiltrating lymphocytes (TILs)”, “succinate dehydrogenase complex subunit A (SDHA)” in the **Introduction** section were also revised accordingly.

References:

1. Palayoor ST, Youmell MY, Calderwood SK, Coleman CN, Price BD. Constitutive activation of IkappaB kinase alpha and NF-kappaB in prostate cancer cells is inhibited by ibuprofen. *Oncogene*. **18**, 7389-94 (1999).
2. Zhang NY, Wang TH, Chou CH, Wu KC, Yang CR, Kung FL, Lin CJ. Ibuprofen treatment ameliorates memory deficits in rats with collagen-induced arthritis by normalizing aberrant MAPK/NF-κB and glutamatergic pathways. *Eur J Pharmacol*. **933**, 175256 (2022).
3. Bomfim CCB, Fisher L, Amaral EP, Mittereder L, McCann K, Correa AAS, Namasivayam S et al. *Mycobacterium tuberculosis* Induces *Irg1* in Murine Macrophages by a Pathway Involving Both TLR-2 and STING/IFNAR Signaling and Requiring Bacterial Phagocytosis. *Front Cell Infect Microbiol*. **12**, 862582 (2022).
4. Peace, C. G. & O'Neill, L. A. The role of itaconate in host defense and inflammation. *The Journal of clinical investigation* **132** (2022).
5. Mills, E. L., Ryan, D. G., Prag, H. A., Dikovskaya, D., Menon, D., Zaslona, Z., Jedrychowski, M. P., Costa, A. S. H., Higgins, M., Hams, E. *et al.* Itaconate is an anti-inflammatory metabolite that activates Nrf2 via alkylation of KEAP1. *Nature* **556**, 113-117 (2018).
6. Wang N, Jiang X, Zhang S, Zhu A, Yuan Y, Xu H, Lei J et al. Structural basis of human monocarboxylate transporter 1 inhibition by anti-cancer drug candidates. *Cell*. **184**, 370-383 (2021).
7. Gudgeon N, Munford H, Bishop EL, Hill J, Fulton-Ward T, Bending D, Roberts J et al. Succinate uptake by T cells suppresses their effector function via inhibition of mitochondrial glucose oxidation. *Cell Rep*. **40**, 111193 (2022).
8. Cervera AM, Bayley JP, Devilee P, McCreath KJ. Inhibition of succinate dehydrogenase dysregulates histone modification in mammalian cells. *Mol Cancer*. **8**, 89 (2009).
9. Xiao M, Yang H, Xu W, Ma S, Lin H, Zhu H, Liu L et al. Inhibition of α-KG-dependent histone and DNA demethylases by fumarate and succinate that are accumulated in mutations of FH and SDH tumor suppressors. *Genes Dev*. **26**, 1326-38 (2012).
10. Hoekstra AS, de Graaff MA, Briaire-de Bruijn IH, Ras C, Seifar RM, van Minderhout I, Cornelisse CJ et al. Inactivation of SDH and FH cause loss of 5hmC and increased H3K9me3 in

- paraganglioma/pheochromocytoma and smooth muscle tumors. *Oncotarget*. **6**, 38777-88 (2015).
11. Janzer A, Stamm K, Becker A, Zimmer A, Buettner R, Kirfel J. The H3K4me3 histone demethylase Fbx110 is a regulator of chemokine expression, cellular morphology, and the metabolome of fibroblasts. *J Biol Chem*. **7**, 30984-92 (2012).
 12. Li ST, Huang D, Shen S, Cai Y, Xing S, Wu G, Jiang Z et al. Myc-mediated SDHA acetylation triggers epigenetic regulation of gene expression and tumorigenesis. *Nat Metab*. **2**, 256-269 (2020).

REVIEWERS' COMMENTS

Reviewer #1 (Remarks to the Author):

The authors have addressed the comments from the reviewers.

Reviewer #2 (Remarks to the Author):

The authors address our concerns and it is a well done study!!!

Point-by-point response to the comments of the Reviewers

Reviewers Comments

Reviewer #1 (Remarks to the Author):

The authors have addressed the comments from the reviewers.

Response: We appreciate the reviewer's positive comments.

Reviewer #2 (Remarks to the Author):

The authors address our concerns and it is a well done study!!!

Response: We appreciate the reviewer's positive comments.